# Learning-Augmented Dynamic Submodular Maximization

**Arpit Agarwal**
Indian Institute of Technology Bombay
`aarpit@iitb.ac.in`

**Eric Balkanski**
Columbia University
`eb3224@columbia.edu`

## Abstract

In dynamic submodular maximization, the goal is to maintain a high-value solution over a sequence of element insertions and deletions with a fast update time. Motivated by large-scale applications and the fact that dynamic data often exhibits patterns, we ask the following question: can predictions be used to accelerate the update time of dynamic submodular maximization algorithms?

We consider the model for dynamic algorithms with predictions where predictions regarding the insertion and deletion times of elements can be used for preprocessing. Our main result is an algorithm with an $O(\text{poly}(\log \eta, \log w, \log k))$ amortized update time over the sequence of updates that achieves a $1/2 - \epsilon$ approximation for dynamic monotone submodular maximization under a cardinality constraint $k$, where the prediction error $\eta$ is the number of elements that are not inserted and deleted within $w$ time steps of their predicted insertion and deletion times. This amortized update time is independent of the length of the stream and instead depends on the prediction error.

## 1 Introduction

Submodular functions are a well-studied family of functions that satisfy a natural diminishing returns property. Since many fundamental objectives are submodular, including coverage, diversity, and entropy, submodular optimization algorithms play an important role in machine learning [18, 8], network analysis [17], and mechanism design [31]. For the canonical problem of maximizing a monotone submodular function under a cardinality constraint, the celebrated greedy algorithm achieves a $1-1/e$ approximation guarantee [27], which is the best approximation guarantee achievable by any polynomial-time algorithm [26]. Motivated by the highly dynamic nature of applications such as influence maximization in social networks and recommender systems on streaming platforms, a recent line of work has studied the problem of dynamic submodular maximization [20, 25, 28, 10, 11, 6, 7]. In the dynamic setting, the input consists of a stream of elements that are inserted or deleted from the set of active elements, and the goal is to maintain, throughout the stream, a subset of the active elements that maximizes a submodular function.

The standard worst-case approach to analyzing the update time of a dynamic algorithm is to measure its update time over the worst sequence of updates possible. However, in many application domains, dynamic data is not arbitrary and often exhibits patterns that can be learned from historical data. Very recent work has studied dynamic problems in settings where the algorithm is given as input predictions regarding the stream of updates [14, 22, 9]. This recent work is part of a broader research area called learning-augmented algorithms (or algorithms with predictions). In learning-augmented algorithms, the goal is to design algorithms that achieve an improved performance guarantee when the error of the prediction is small and a bounded guarantee even when the prediction error is arbitrarily large. A lot of the effort in this area has been focused on using predictions to improve the competitive ratio of online algorithms (see, e.g., [23, 30, 32, 24, 12, 3, 4, 19, 15, 5]), and more generally to improve the solution quality of algorithms.

38th Conference on Neural Information Processing Systems (NeurIPS 2024).

For dynamic submodular maximization with predictions, Liu and Srinivas [22] considered a predicted-deletion model and achieved, under some mild assumption, a $0.3178$ approximation and an $\tilde{O}(\text{poly}(k, \log n))^1$ update time for dynamic monotone submodular maximization under a matroid constraint of rank $k$ and over a stream of length $n$. This approximation is an improvement over the best-known $1/4$ approximation for dynamic monotone submodular maximization (without predictions) under a matroid constraint with an update time that is sublinear in $n$ [7, 11]. Since the update time of dynamic algorithms is often the main bottleneck in large-scale problems, another promising direction is to leverage predictions to improve the update time of dynamic algorithms.

*Can predictions help to accelerate the update time of dynamic submodular maximization algorithms?*

We note that the three very recent papers on dynamic algorithms with predictions have achieved improved update times for several dynamic graph problems [14, 22, 9]. However, to the best of our knowledge, there is no previous result that achieves an improved update time for dynamic submodular maximization by using predictions.

**Our contributions.** In dynamic submodular maximization, the input is a submodular function $f : 2^V \to \mathbb{R}_{\geq 0}$ and a sequence of $n$ element insertions and deletions. The active elements $V_t \subseteq V$ at time $t$ are the elements that have been inserted and have not yet been deleted during the first $t$ updates. The goal is to maintain, at every time step $t$, a solution $S_t \subseteq V_t$ that is approximately optimal with respect to $V_t$ while minimizing the number of queries to $f$ at each time step, which is referred to as the update time. As in [14, 9], we consider a prediction model where, at time $t = 0$, the algorithm is given a prediction regarding the sequence of updates, which can be used for preprocessing. More precisely, at time $t = 0$, the algorithm is given predictions $(\hat{t}_a^+, \hat{t}_a^-)$ about the insertion and deletion time of each element $a$. A dynamic algorithm with predictions consists of two phases. During the precomputation phase at $t = 0$, the algorithm uses the predictions to perform queries before the start of the stream. During the streaming phase at time steps $t > 0$, the algorithm performs queries, and uses the precomputations, to maintain a good solution with respect to the true stream.

In this model, there is a trivial algorithm that achieves a constant update time when the predictions are exactly correct and an $O(u)$ update time when the predictions are arbitrarily wrong. Here, $u$ is the update time of an arbitrary algorithm $\mathcal{A}$ for the problem without predictions. This algorithm precomputes, for each future time step $t$, a solution for the elements that are predicted to be active at time $t$ and then, during the streaming phase, returns the precomputed solutions while the prediction is correct and switches to running algorithm $\mathcal{A}$ at the first error in the prediction. Thus, the interesting question is whether it is possible to obtain an improved update time not only when the predictions are exactly correct, but more generally when the error in the predictions is small. An important component of our model is the measure for the prediction error. Given a time window tolerance $w$, an element $a$ is considered to be correctly predicted if the predicted insertion and deletion times of $a$ are both within $w$ times steps of its true insertion and deletion times. The prediction error $\eta$ is then the number of elements that are not correctly predicted. Thus, $\eta = 0$ if the predictions are exactly correct and $\eta = \Theta(n)$ if the predictions are completely wrong.

For dynamic monotone submodular maximization (without predictions) under a cardinality constraint $k$, Lattanzi et al. [20] and Monemizadeh [25] concurrently obtained dynamic algorithms with $O(\text{poly}(\log n, \log k))$ and $O(\text{polylog}(n) \cdot k^2)$ amortized update time, respectively, that achieve a $1/2 - \epsilon$ approximation. More recently, Banihashem et al. [7] achieved a $1/2 - \epsilon$ approximation with a $O(k \cdot \text{polylog}(k))$ amortized update time. Our main result is the following.

**Theorem.** *For monotone submodular maximization under a cardinality constraint $k$, there is a dynamic algorithm with predictions that, for any tolerance $w$ and constant $\epsilon > 0$, achieves an amortized expected query complexity per update of $O(\text{poly}(\log \eta, \log w, \log k))$, an approximation of $1/2 - \epsilon$ in expectation, and a query complexity of $\tilde{O}(n)$ during the precomputation phase.*

We note that, when the prediction error $\eta$ is arbitrarily large, our algorithm matches the $O(\text{poly}(\log n, \log k))$ amortized expected query complexity per update in [20]. It also achieves an approximation that matches the optimal approximation for dynamic algorithms (without predictions) with update time that is sublinear in $n$. An intriguing open question is whether an improvement in update time can be obtained in the predicted-deletion model of [22] with no preprocessing and instead a predicted deletion time for each element $a$ is given at the time when $a$ is inserted.

---

[1]In this paper, we use the notation $\tilde{O}(g(n))$ as shorthand for $O(g(n) \log^k g(n))$.

**Related work.** For monotone submodular maximization under a cardinality constraint, dynamic algorithms with $O(\text{poly}(\log n, \log k))$ and $O(\text{polylog}(n) \cdot k^2)$ amortized update time that achieve a $1/2$ approximation were concurrently obtained in [20, 25]. Recently, Banihashem et al. [7] gave a $1/2$ approximation algorithm with a $O(k \cdot \text{polylog}(k))$ amortized update time. Chen and Peng [10] showed that any dynamic algorithm with an approximation better than $1/2$ must have $\text{poly}(n)$ amortized query complexity per update. For matroid constraints, Chen and Peng [10] obtained an insertion-only algorithm. As mentioned in [29], the streaming algorithm in [13] can be adapted to also give an insertion-only algorithm. Two $1/4$-approximation dynamic algorithms with $\tilde{O}(k)$ and $\tilde{O}(\text{polylog}(n) \cdot k^2)$ amortized update time were concurrently obtained in [7] and [11].

Algorithms with predictions have been studied in a wide range of areas, including online algorithms [23, 30], mechanism design [1, 33], and differential privacy [2]. Improved update times for several dynamic graph problems were very recently obtained by leveraging predictions [14, 22, 9]. In particular, Liu and Srinivas [22] obtained, under some mild assumption on the prediction error, a $0.3178$ approximation and a $\tilde{O}(\text{poly}(k, \log n))$ update time for dynamic monotone submodular maximization under a matroid constraint of rank $k$ in the more challenging predicted-deletion model. Thus, by using predictions, this result improves the $1/4$ approximation achieved, without predictions, in [7] and [11] (but does not improve the $1/2$ approximation for cardinality constraints). The results in [22] use a framework that takes as input an insertion-only dynamic algorithm. In contrast, we develop a framework that uses a fully dynamic algorithm and a deletion-robust algorithm.

## 2 Preliminaries

A function $f : 2^V \to \mathbb{R}$ defined over a ground set $V$ is submodular if for all $S \subseteq T \subseteq V$ and $a \in V \setminus T$, we have that $f_S(a) \geq f_T(a)$, where $f_S(a) = f(S \cup \{a\}) - f(S)$ is the marginal contribution of $a$ to $S$. It is monotone if $f(S) \leq f(T)$ for all $S \subseteq T \subseteq V$. We consider the canonical problem of maximizing a monotone submodular function $f$ under a cardinality constraint $k$.

In **dynamic submodular maximization**, there is a stream $\{(a_t, o_t)\}_{t=1}^n$ of $n$ element insertions and deletions where $o_t \in \{\text{insert}, \text{deletion}\}$ and $a_t$ is an element in $V$. The active elements $V_t$ are the elements that have been inserted and have not been deleted by time $t$. We assume that $(a_t, \text{insertion})$ and $(a_t, \text{deletion})$ can occur in the stream only if $a_t \notin V_t$ and $a_t \in V_t$, respectively, and that each element is inserted at most once.[2] The goal is to maintain a solution $S_t \subseteq V_t$ that approximates the optimal solution over $V_t$, which we denote by $O_t$. Since our algorithmic framework takes as input a dynamic algorithm, we formally define dynamic algorithms in terms of black-box subroutines that are used in our algorithms.

**Definition 1.** *A dynamic algorithm* DYNAMIC$(f, k)$ *consists of the following four subroutines to process a stream* $\{(a_t, o_t)\}_{t=1}^n$. DYNAMICINIT$(f, k)$ *initializes a data structure $A$ at $t = 0$. If $o_t = $ insertion or $o_t = $ deletion,* DYNAMICINS$(A, a_t)$ *or* DYNAMICDEL$(A, a_t)$ *insert in $A$ or delete from $A$ element $a_t$ at time $t$. At time $t$,* DYNAMICSOL$(A)$ *returns $S_t \subseteq V(A)$ s.t. $|S| \leq k$, where $V(A) = V_t$ is the set of elements that have been inserted in and not been deleted from $A$.*

A dynamic algorithm achieves an $\alpha$-approximation in expectation if, for all time steps $t$, $\mathbf{E}[f(S_t)] \geq \alpha \cdot \max_{S \subseteq V_t : |S| \leq k} f(S)$ and has a $u(n, k)$ amortized expected query complexity per update if its expected total number of queries is $n \cdot u(n, k)$.

In **dynamic submodular maximization with predictions**, the algorithm is given at time $t = 0$ predictions $\{(\hat{t}_a^+, \hat{t}_a^-)\}_a$ about the insertion and deletion time of elements $a$. The prediction error $\eta$ is the number of elements that are incorrectly predicted, where an element $a$ is correctly predicted if it is inserted and deleted within a time window, of size parameterized by a time window tolerance $w$, that is centered at the time at which $a$ is predicted to be inserted and deleted.

**Definition 2.** *Given a tolerance $w \in \mathbb{Z}_+$, predictions $\{(\hat{t}_a^+, \hat{t}_a^-)\}_{a \in V}$, and true insertion and deletions times $\{(t_a^+, t_a^-)\}_{a \in V}$, the prediction error is $\eta = |\{a \in V : |\hat{t}_a^+ - t_a^+| > w \text{ or } |\hat{t}_a^- - t_a^-| > w\}|$.*

We note that $\eta = w = 0$ corresponds to the predictions being exactly correct and $\eta = O(n)$ and $w = O(n)$ corresponds to thems being arbitrarily wrong. We also emphasize that our model does not require knowing the entire ground set $V$ at $t = 0$. Elements that are not known at $t = 0$ are assumed to have predicted arrival and departure times equal to infinity, and contribute to the prediction error $\eta$.

---

[2] If an element $a$ is re-inserted, a copy $a'$ of $a$ can be created.

**Deletion-robust submodular maximization.** Our framework also takes as input a *deletion-robust algorithm*, which we formally define in terms of black-box subroutines. A deletion-robust algorithm finds a solution $S \subseteq V$ that is robust to the deletion of at most $d$ elements.

**Definition 3.** *Given a function $f : 2^V \rightarrow \mathbb{R}$, a cardinality constraint $k$, and a maximum number of deletions parameter $d$, a deletion-robust algorithm $\textsc{Robust}(f, V, k, d)$ consists of a first subroutine $\textsc{Robust1}(f, V, k, d)$ that returns a robust set $R \subseteq V$ and a second subroutine $\textsc{Robust2}(f, R, D, k)$ that returns a set $S \subseteq R \setminus D$ such that $|S| \leq k$.*

A deletion-robust algorithm achieves an $\alpha$ approximation if, for any $f$, $V$, $k$, and $d$, the subroutine $\textsc{Robust1}(f, V, k, d)$ returns $R$ such that, for any $D \subseteq V$ such that $|D| \leq d$, $\textsc{Robust2}(f, R, D, k)$ returns $S$ such that $\mathbf{E}[f(S)] \geq \alpha \cdot \max_{T \subseteq V \setminus D : |T| \leq k} f(T)$. Kazemi et al. [16] show that there is a deletion-robust algorithm for monotone submodular maximization under a cardinality constraint such that $\textsc{Robust1}$ returns a set $R$ of size $|R| = O(\epsilon^{-2} d \log k + k)$. It achieves a $1/2 - \epsilon$ approximation in expectation, $\textsc{Robust1}$ has $O(|V| \cdot (k + \epsilon^{-1} \log k))$ query complexity, and $\textsc{Robust2}$ has $O\left((\epsilon^{-2} d \log k + k) \cdot \epsilon^{-1} \log k\right)$ query complexity.

**The algorithmic framework.** We present an algorithmic framework that decomposes dynamic algorithms with predictions into two subroutines, $\textsc{Precomputations}$ and $\textsc{UpdateSol}$. The remainder of the paper then consists of designing and analyzing these subroutines. We first introduce some terminology. An element $a$ is said to be correctly predicted if $|\hat{t}_a^+ - t_a^+| \leq w$ and $|\hat{t}_a^- - t_a^-| \leq w$. The *predicted elements* $\hat{V}_t$ consist of all elements that could potentially be active at time $t$ if correctly predicted, i.e., the elements $a$ such that $\hat{t}_a^+ \leq t + w$ and $\hat{t}_a^- \geq t - w$. During the precomputation phase, the first subroutine, $\textsc{Precomputations}$, takes as input the predicted elements $\hat{V}_t$ and outputs, for each time step $t$, a data structure $P_t$ that will then be used at time $t$ of the streaming phase to compute a solution efficiently. During the streaming phase, the active elements $V_t$ are partitioned into the *predicted active elements* $V_t^1 = V_t \cap \hat{V}_t$ and the *unpredicted active elements* $V_t^2 = V_t \setminus \hat{V}_t$. The second subroutine, $\textsc{UpdateSol}$, is given $V_t^1$ and $V_t^2$ as input and computes a solution $S \subseteq V_t^1 \cup V_t^2$ at each time step. $\textsc{UpdateSol}$ is also given as input precomputations $P_t$ and the current prediction error $\eta_t$. It also stores useful information for future time steps in a data structure $A$.

---

**Algorithm 1** The Algorithmic Framework

---

    **Input:** function $f : 2^V \rightarrow \mathbb{R}$, constraint $k$, predictions $\{(\hat{t}_a^+, \hat{t}_a^-)\}_{a \in V}$, tolerance $w$

1:  $\hat{V}_t \leftarrow \{a \in V : \hat{t}_a^+ \leq t + w \text{ and } \hat{t}_a^- \geq t - w\}$ for $t \in [n]$
2:  $\{P_t\}_{t=1}^n \leftarrow \textsc{Precomputations}(f, \{\hat{V}_i\}_{i=1}^n, k)$
3:  $V_0, A \leftarrow \emptyset$
4:  **for** $t = 1$ to $n$ **do**
5:      Update active elements $V_t$ according to operation at time $t$
6:      $V_t^1 \leftarrow V_t \cap \hat{V}_t$, $V_t^2 \leftarrow V_t \setminus \hat{V}_t$
7:      $\eta_t \leftarrow$ current prediction error
8:      $A, S \leftarrow \textsc{UpdateSol}(f, k, A, t, P_t, V_t^1, V_t^2, \hat{V}_t, \eta_t)$
9:      **return** $S$

---

## 3 The warm-up algorithm

In this section, we present subroutines that achieve an $\tilde{O}(\eta + w + k)$ amortized update time and a $1/4 - \epsilon$ approximation in expectation. These warm-up subroutines assume that the error $\eta$ is known. They take as input a dynamic algorithm (without predictions) $\textsc{Dynamic}$ and a deletion-robust algorithm $\textsc{Robust}$ algorithm. The proofs are all deferred to the appendix.

**The precomputations subroutine** A main observation is that the problem of finding a solution $S^1 \subseteq V_t^1$ among the predicted active elements corresponds to a deletion-robust problem over $\hat{V}_t$ where the deleted elements $D$ are the predicted elements $\hat{V}_t \setminus V_t$ that are not active at time $t$. $\textsc{WarmUp-Precomputations}$ thus calls, for each time $t$, the first stage $\textsc{Robust1}$ of $\textsc{Robust}$,

$$\textsc{WarmUp-Precomputations}(f, \{\hat{V}_i\}_{i=1}^n, k) = \{\textsc{Robust1}(f, \hat{V}_t, k, d = \eta + 2w)\}_{t=1}^n.$$

The algorithm sets the maximum number of deletions parameter $d$ for ROBUST1 to $\eta + 2w$ because the number of predicted elements $\hat{V}_t \setminus V_t$ that are not active at time $t$ is at most $\eta + 2w$ (Lemma 8).

**The updatesol subroutine** WARMUP-UPDATESOL finds a solution $S^1 \subseteq V_t^1$ by calling ROBUST2 over the precomputed $P_t$ and deleted elements $D = \hat{V}_t \setminus V_t$. To find a solution $S^2 \subseteq V_t^2$ among the unpredicted active elements, we use DYNAMIC over the stream of element insertions and deletions that result in unpredicted active elements $V_1^2, \ldots, V_n^2$, which is the stream that inserts elements $V_t^2 \setminus V_{t-1}^2$ and deletes elements $V_{t-1}^2 \setminus V_t^2$ at time $t$. The solution $S^2$ is then the solution produced by DYNAMICSOL over this stream. The solution $S$ returned by UPDATESOL is the best solution between $S^1$ and $S^2$.

---

**Algorithm 2** WARMUP-UPDATESOL

    **Input:** function $f$, constraint $k$, data structure $A$, time $t$, precomputations $P_t$, predicted active elements $V_t^1$, unpredicted active elements $V_t^2$, predicted elements $\hat{V}_t$

1: $S^1 \leftarrow$ ROBUST2$(f, P_t, \hat{V}_t \setminus V_t^1, k)$
2: **if** $t = 1$ **then** $A \leftarrow$ DYNAMICINIT$(f, k)$
3: **for** $a \in V_t^2 \setminus V(A)$ **do** DYNAMICINS$(A, a)$
4: **for** $a \in V(A) \setminus V_t^2$ **do** DYNAMICDEL$(A, a)$
5: $S^2 \leftarrow$ DYNAMICSOL$(A)$
6: **return** $A, \arg\max\{f(S^1), f(S^2)\}$

---

**The analysis of the warm-up algorithm** We first analyze the approximation. We let $\alpha_1$ and $\alpha_2$ denote the approximations achieved by ROBUST and DYNAMIC. The first lemma shows that solution $S^1$ is an $\alpha_1$ approximation to the optimal solution over the predicted active elements $V_t^1$ and that solution $S^2$ is an $\alpha_2$ approximation to the optimal solution over the unpredicted active elements $V_t^2$.

**Lemma 1.** *At every time step $t$, $\mathbf{E}[f(S^1)] \geq \alpha_1 \cdot \text{OPT}(V_t^1)$ and $\mathbf{E}[f(S^2)] \geq \alpha_2 \cdot \text{OPT}(V_t^2)$.*

The main lemma for the amortized query complexity bounds the number of calls to DYNAMICINS.

**Lemma 2.** WARMUP-UPDATESOL *makes at most $2\eta$ calls to* DYNAMICINS *on $A$ over the stream.*

The main result for the warm-up algorithm is the following.

**Theorem 1.** *For monotone submodular maximization under a cardinality constraint $k$, Algorithm 1 with the* WARMUP-PRECOMPUTATIONS *and* WARMUP-UPDATESOL *subroutines achieves, for any tolerance $w$ and constant $\epsilon > 0$, an amortized expected query complexity per update during the streaming phase of $\tilde{O}(\eta + w + k)$, an approximation of $1/4 - \epsilon$ in expectation, and a query complexity of $\tilde{O}(n^2 k)$ during the precomputation phase.*

In the next sections, we improve the dependencies on $\eta, w$, and $k$ for the query complexity per update from linear to logarithmic, the approximation from $1/4$ to $1/2$, and the precomputations query complexity from $O(n^2 k)$ to $\tilde{O}(n)$. We also remove the assumption that the prediction error is known.

## 4 The UpdateSol subroutine

In this section, we improve the dependencies in $\eta, w$, and $k$ for the amortized query complexity from linear to logarithmic, which is the main technical challenge. For finding a solution over the predicted active elements $V_t^1$, the main idea is to not only use precomputations $P_t$, but also to exploit computations from previous time steps $t' < t$ over the previous predicted active elements $V_{t'}^1$. As in the warm-up subroutine, the new UPDATESOLMAIN subroutine also uses a precomputed deletion-robust solution $P_t$, but it requires $P_t$ to satisfy a property termed the *strongly robust property* (Definition 4 below), which is stronger than the deletion-robust property of Definition 3. A strongly robust solution comprises two components $Q$ and $R$, where $R$ is a small set of elements that have a high marginal contribution to $Q$. The set $Q$ is such that, for any deleted set $D$, $f(Q \setminus D)$ is guaranteed to, in expectation over the randomization of $Q$, retain a large amount of $f(Q)$.

**Definition 4.** *A pair of sets $(Q, R)$ is $(d, \epsilon, \gamma)$-strongly robust, where $d, k, \gamma \geq 0$ and $\epsilon \in [0, 1]$, if*

- **Size.** $|Q| \leq k$ and $|R| = O(\epsilon^{-2}(d+k)\log k)$ *with probability* 1,

- **Value.** $f(Q) \geq |Q|\gamma/(2k)$ *with probability* 1. *In addition, if* $|Q| < k$, *then for any set* $S \subseteq V \setminus R$ *we have* $f_Q(S) < |S|\gamma/(2k) + \epsilon\gamma$.

- **Robustness.** *For any* $D \subseteq V$ *s.t.* $|D| \leq d$, $\mathbf{E}_Q[f(Q \setminus D)] \geq (1-\epsilon)f(Q)$.

The set $P$ returned by the first stage ROBUST1$(f, V, k, d)$ of the deletion-robust algorithm of Kazemi et al. [16] can be decomposed into two sets $Q$ and $R$ that are, for any $d, \epsilon > 0$ and with $\gamma = \mathrm{OPT}(V)$, where $\mathrm{OPT}(V) := \max_{S \subseteq V:|S| \leq k} f(S)$, $(d, \epsilon, \gamma)$-strongly robust.[3] Thus, with the ROBUST algorithm of [16], the set $P_t$ returned by WARMUP-PRECOMPUTATIONS can be decomposed into $P_t$ and $Q_t$ that are, for any $\epsilon > 0$, $(2(\eta+2w), \epsilon, \mathrm{OPT}(\hat{V}_t))$-strongly robust. We omit the proof of the $(d, \epsilon, \gamma)$-strongly robust property of ROBUST1 from [16] and, in the next section, we instead prove strong-robustness for our PRECOMPUTATIONSMAIN subroutine which has better overall query complexity than [16].

The UPDATESOLMAIN subroutine proceeds in phases. During each phase, UPDATESOLMAIN maintains a data structure $(B, A, \eta_{\mathrm{old}})$. The set $B = Q_{t'}$ is a fixed base set chosen during the first time step $t'$ of the current phase. $A$ is a dynamic data structure used by a dynamic submodular maximization algorithm DYNAMIC that initializes $A$ over function $f_B$, cardinality constraint $k - |B|$, and a parameter $\gamma$ to be later discussed. If a new phase starts at time $t$, note that if $(Q_t, R_t)$ are strongly robust, then the only predicted active elements $V_t^1$ that are needed to find a good solution at time $t$ are, in addition to $B = Q_t$, the small set of elements $R_t$ that are also in $V_t^1$. Thus to find a good solution for the function $f_B$, $(R_t \cap V_t^1) \cup V_t^2$ are inserted into $A$. The solution that UPDATESOLMAIN outputs at time step $t$ are the active elements $B \cap V_t$ that are in the base for the current phase, together with the solution DYNAMICSOL$(A)$ maintained by $A$.

---

**Algorithm 3** UPDATESOLMAIN

---

**Input:** function $f$, data structure $(B, A, \eta_{\mathrm{old}})$, constraint $k$, precomputations $P_t = (Q_t, R_t)$, $t$, upper bound $\eta_t'$ on prediction error, $V_t^1$, $V_t^2$, $V_{t-1}$, parameter $\gamma_t$

1: **if** $t = 1$ or $|\mathrm{Ops}^\star(A)| > \frac{\eta_{\mathrm{old}}}{2} + w$ **then**                  ▷ Start a new phase
2:     $B \leftarrow Q_t$
3:     $A \leftarrow \mathrm{DYNAMICINIT}(f_B, k - |B|, \gamma = \gamma_t(k - |B|)/k)$
4:     **for** $a \in (R_t \cap V_t^1) \cup V_t^2$ **do** DYNAMICINS$(A, a)$
5:     $\eta_{\mathrm{old}} \leftarrow \eta_t'$
6: **else**                                                                              ▷ Continue the current phase
7:     **for** $a \in V_t \setminus V_{t-1}$ **do** DYNAMICINS$(A, a)$
8:     **for** $a \in (\mathrm{ELEM}(A \cap V_{t-1}) \setminus V_t$ **do** DYNAMICDEL$(A, a)$
9: $S \leftarrow (B \cup \mathrm{DYNAMICSOL}(A)) \cap V_t$
10: **return** $(B, A, \eta_{\mathrm{old}}), S$

---

During the next time steps $t$ of the current phase, if an element $a$ is inserted into the stream then $a$ is inserted in $A$ (independently of the predictions). If an element is deleted from the stream, then if it was in $A$, it is deleted from $A$. We define $\mathrm{Ops}^\star(A)$ to be the collection of all insertion and deletion operations to $A$, excluding the insertions of elements in $(R_t \cap V_t^1) \cup V_t^2$ at the time $t$ where $A$ was initialized. The current phase ends when $\mathrm{Ops}^\star(A)$ exceeds $\eta_{\mathrm{old}}/2 + w$. Since the update time of the dynamic algorithm in [20] depends on length of the stream, we upper bound the length of the stream handled by $A$ during a phase.

**The approximation**    The parameter $\gamma_t$ corresponds to a guess for $\mathrm{OPT}_t := \mathrm{OPT}(V_t)$. In Section 6, we present the UPDATESOLFULL subroutine which calls UPDATESOLMAIN with different guesses $\gamma_t$. This parameter $\gamma_t$ is needed when initializing DYNAMIC because our analysis requires that DYNAMIC satisfies a property that we call threshold-based, which we formally define next.

**Definition 5.** *A dynamic algorithm* DYNAMIC *is threshold-based if, when initialized with threshold parameter* $\gamma$ *such that* $\gamma \leq \mathrm{OPT}_t \leq (1+\epsilon)\gamma$, *a cardinality constraint* $k$, *and* $\epsilon > 0$, *it maintains a data structure* $A_t$ *and solution* $\mathrm{SOL}_t = \mathrm{DYNAMICSOL}(A_t)$ *that satisfy, for all* $t$, $f(\mathrm{SOL}_t) \geq \frac{\gamma}{2k}|\mathrm{SOL}_t|$ *and, if* $|\mathrm{SOL}_t| < (1-\epsilon)k$, *then for any set* $S \subseteq V(A_t)$, *we have* $f_{\mathrm{SOL}_t}(S) < \frac{|S|\gamma}{2k} + \epsilon\gamma$.

---

[3]The size of $R$ in [16] is $O(d \log k/\epsilon)$ which is better than what is required to be strongly robust.

**Lemma 3.** *The* DYNAMIC *algorithm of Lattanzi et al. [20][4] is threshold-based.*

The main lemma for the approximation guarantee is the following.

**Lemma 4.** *Consider the data structure* $(B, A, \eta_{old})$ *returned by* UPDATESOLMAIN *at time* $t$. *Let* $t'$ *be the time at which* $A$ *was initialized,* $(Q_{t'}, R_{t'})$ *and* $\gamma_{t'}$ *be the precomputations and guess for* OPT$_{t'}$ *inputs to* UPDATESOLMAIN *at time* $t'$. *If* $(Q_{t'}, R_{t'})$ *are* $(d = 2(\eta_{old} + 2w), \epsilon, \gamma_{t'})$-*strongly robust,* $\gamma_{t'}$ *is such that* $\gamma_{t'} \leq$ OPT$_t \leq (1 + \epsilon)\gamma_{t'}$, *and* DYNAMIC *is a threshold-based dynamic algorithm, then the set* $S$ *returned by* UPDATESOLMAIN *is such that* $\mathbf{E}[f(S)] \geq \frac{1-5\epsilon}{2}\gamma_{t'}$.

**The update time**   We next analyze the query complexity of UPDATESOLMAIN. Recall that $u(n, k)$ denotes the amortized query complexity per update of DYNAMIC.

**Lemma 5.** *Consider the data structure* $(B, A, \eta_{old})$ *returned by* UPDATESOLMAIN *at time* $t$. *Let* $t'$ *be the time at which* $A$ *was initialized and* $(Q_{t'}, R_{t'})$ *be the precomputations input to* UPDATESOLMAIN *at time* $t'$. *If precomputations* $(Q_{t'}, R_{t'})$ *are* $(d = 2(\eta_{old} + 2w), \epsilon, \gamma)$ *strongly-robust, then the total number of queries performed by* UPDATESOL *during the* $t - t'$ *time steps between time* $t'$ *and time* $t$ *is* $O(u((\eta_{old} + w + k) \log k, k) \cdot (\eta_{old} + w + k) \log k)$. *Additionally, the number of queries between time* $1$ *and* $t$ *is upper bounded by* $O(u(t, k) \cdot t)$.

## 5   The Precomputations subroutine

In this section, we provide a PRECOMPUTATIONS subroutine that has an improved query complexity compared to the warm-up precomputations subroutine. Recall that the warm-up subroutine computes a robust solution over predicted elements $\hat{V}_t$, independently for all times $t$. The improved PRECOMPUTATIONS does not do this independently for each time step. Instead, it relies on the following lemma that shows that the data structure maintained by the dynamic algorithm of [20] can be used to find a strongly robust solution without any additional query.

**Lemma 6.** *Let* DYNAMIC$(\gamma, \epsilon)$ *be the dynamic submodular maximization algorithm of [20] and* $A$ *be the data structure it maintains. There is a* ROBUST1FROMDYNAMIC *algorithm such that, given as input a deletion size parameter* $d$, *and the data structure* $A$ *at time* $t$ *with* $\gamma \leq$ OPT$_t \leq (1 + \epsilon)\gamma$, *it outputs sets* $(Q, R)$ *that are* $(d, \epsilon, \gamma)$-*strongly robust with respect to the ground set* $V_t$. *Moreover, this algorithm does not perform any oracle queries.*

The improved PRECOMPUTATIONSMAIN subroutine runs the dynamic algorithm of [20] and then, using the ROBUST1FROMDYNAMIC algorithm of Lemma 6, computes a strongly-robust set from the data structure maintained by the dynamic algorithm.

---

**Algorithm 4** PRECOMPUTATIONSMAIN

---

    **Input:** function $f : 2^V \rightarrow \mathbb{R}$, constraint $k$, predicted elements $\hat{V}_1, \ldots, \hat{V}_n \subseteq V$, time error tolerance $w$, parameter $\gamma$, parameter $h$
1:  $\hat{A} \leftarrow$ DYNAMICINIT$(f, k, \gamma)$
2:  **for** $t = 1$ to $n$ **do**
3:     **for** $a \in \hat{V}_t \setminus \hat{V}_{t-1}$ **do** DYNAMICINS$(\hat{A}, a)$
4:     **for** $a \in V(\hat{A}) \setminus \hat{V}_t$ **do** DYNAMICDEL$(\hat{A}, a)$
5:     **if** $|V(\hat{A})| > 0$ **then** $Q_t, R_t \leftarrow$ ROBUST1FROMDYNAMIC$(f, \hat{A}, k, 2(h + 2w))$
6:  **return** $\{(Q_t, R_t)\}_{t=1}^n$

---

The parameters $\gamma$ and $h$ correspond to guesses for OPT and $\eta$ respectively.

**Lemma 7.** *The total query complexity of the* PRECOMPUTATIONSMAIN *algorithm is* $n \cdot u(n, k)$, *where* $u(n, k)$ *is the amortized query complexity of calls to* DYNAMIC.

---

[4]Note that the initially published algorithm in [20] had an issue with correctness, we refer to the revised version.

# 6 The full algorithm

The UPDATESOLMAIN and PRECOMPUTATIONSMAIN subroutines use guesses $\gamma_t$ and $h$ for the optimal value $\text{OPT}_t$ at time $t$ and the total prediction error $\eta$. In Appendix D, we describe the full UPDATESOLFULL and PRECOMPUTATIONSFULL subroutines that call the previously defined subroutines over different guesses $\gamma_t$ and $h$. The parameters of these calls must be carefully designed to bound the streaming amortized query complexity per update and precomputations query complexity. By combining the algorithmic framework (Algorithm 1) together with subroutines UPDATESOLFULL and PRECOMPUTATIONSFULL, we obtain our main result.

**Theorem 2.** *Algorithm 1 with subroutines* UPDATESOLFULL *and* PRECOMPUTATIONSFULL *is a dynamic algorithm that, for any tolerance $w$ and constant $\epsilon > 0$, achieves an amortized expected query complexity per update during the streaming phase of $O(\text{poly}(\log \eta, \log w, \log k))$, an approximation of $1/2 - \epsilon$ in expectation, and a query complexity[5] of $\tilde{O}(n)$ during the precomputation phase.*

Note that the query complexity per update during the streaming phase and the query complexity during the precomputation phase both have a polynomial dependence on $\epsilon$. Additionally, note that our update bound is not constant even when the prediction error is 0. However, with the following simple change, our algorithm achieves a constant update time when the predictions are exactly correct, while also maintaining its current guarantees: (1) as additional precomputations, also compute a predicted solution $S_t$ for each time $t$ assuming the predictions are exactly correct, (2) during the streaming phase, as long as the predictions are exactly correct, return the precomputed predicted solution $S_t$. At the first time step where the predictions are no longer exactly correct, switch to our main algorithm in the paper.

# 7 Experiments

## 7.1 Experimental Setup

**Benchmarks.** We compare our **DYNAMICWPRED** algorithm to two benchmarks. The first is the **DYNAMIC** submodular maximization algorithm of [20], which does not use predictions. The second is the **OFFLINEGREEDY** algorithm that computes an offline greedy solution $\hat{S}_t$ at each time step $t$ based on the set $\hat{V}_t$ of available items in the predicted stream. In the streaming phase, it outputs the solution $\hat{S}_t \cap V_t$ at time $t$ based on the available items $V_t$. Note that this algorithm does not make any queries other than the queries used for pre-computation. These benchmarks are at two extremes in terms of their reliance on the predictions. DYNAMICWPRED uses the DYNAMIC algorithm of [20] and the ROBUST algorithm of [16] as subroutines. We implemented our algorithm and OFFLINEGREEDY in C++, and used the C++ implementation of DYNAMIC that is provided by [20]. We set $\epsilon = 0.2$ for all algorithms.

**Metrics.** We report the total number of oracle calls made by each algorithm when processing the actual stream $(\mathbf{a}, \mathbf{o})$. This does not include any oracle calls during the pre-computation phase. We also report the average function value $\frac{1}{n}\sum_{t\in[n]} f(S_t)$ of the output sets over time steps $t \in [n]$. Each experiment is repeated 5 times and the average values are reported.

**Datasets and submodular function.** We perform experiments on a subset of the Enron dataset from the SNAP Large Networks Data Collection [21]. We select a set $V$ of 200 nodes from the graph, and consider the subgraph induced by $V \cup N(V)$ where $N(V)$ is the set of neighboring nodes of $V$. This resulted in a subgraph with 7845 vertices and 20033 edges. The submodular function is the dominating set objective function over the ground set $V$ with $|V| = 200$. Specifically, for a subset of nodes $S \subseteq V$, we define $f(S) = |N(S) \cup S|$, where $N(S)$ is the set of neighboring nodes of $S$. This function is monotone and submodular.

---

[5]We note that, despite the amortized query complexity of Lattanzi et al. [20] being in expectation, the asymptotic bound on the precomputation query complexity can hold deterministically, instead of in expectation, by forcing PRECOMPUTATIONSFULL to terminate if it has performed a number of queries that is larger than $\epsilon^{-1}$ times its expected number of queries (note that the precomputation query complexity only depends on known parameters, $n$ and $k$). By Markov's inequality, such an early termination happens with probability at most $\epsilon$. Thus, even with no guarantees on the approximation achieved in these early termination cases, the loss in the expected approximation caused by this forced termination is at most $1 - \epsilon$.

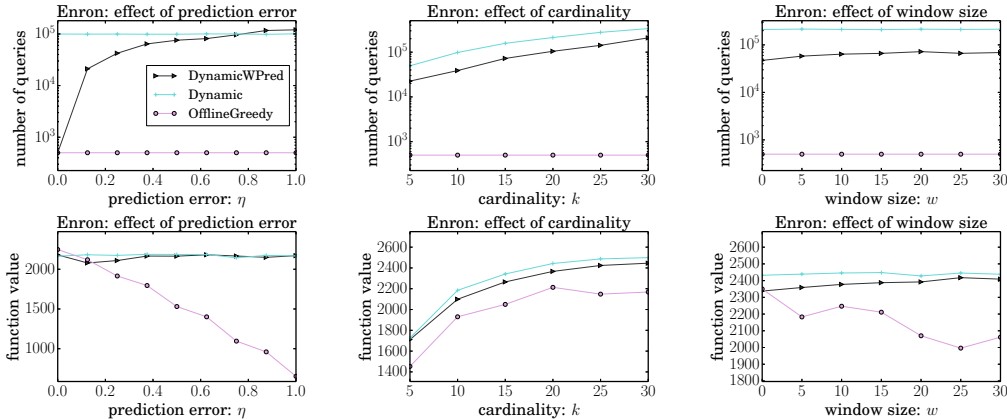

Figure 1: The number of queries and function value of our algorithm, DYNAMICWPRED, and the two benchmarks for the Enron data and sliding window stream with $l = 50$ as a function of the prediction error $\eta$ with $k = 10$ and $w = 0$ (column 1), as a function of the cardinality parameter $k$ with $\eta = 0.25$ and $w = 0$ (column 2), and as a function of the window size parameter $w$ with $k = 20$ and $\eta = 0.125$ (column 3).

**Generation of true stream.** We consider the sliding window protocol from [20] in order to generate the dynamic stream of insertions and deletions. Specifically, we process the nodes $V$ in an arbitrary order and consider a sliding window of size $l$. When the window reaches a node $a$, we add the operation $(a, \text{insert})$ to the dynamic stream. Similarly, after $l$ steps when the node $a$ leaves the window, we add the operation $(a, \text{delete})$ to the dynamic stream. Since, $|V| = 200$, we have $n = 400$ for all experiments. We report results with $l = 50$ (observations were similar for other values of $l$).

**Generation of predicted stream.** Given a target prediction error $\eta$ and window size $w$, we generate the predicted stream for our experiments by adding perturbations to the actual stream such that Definition 2 is satisfied. Note that we add these perturbations while maintaining the consistency of the stream, i.e. the insertion of an element always happens before its deletion. In particular, we first select a set $E$ of $\eta/2$ elements uniformly at random and let $\mathcal{T}$ denote the set of all insertion and deletion times of these elements in the actual stream. For each $e \in E$, we assign new insertion and deletion times in the predicted stream by randomly drawing from $\mathcal{T}$. We make sure that these new insertion and deletion times are consistent and are at least a distance of $w$ from the corresponding old times. The insertion and deletion times for $e \notin E$ remain the same so far. Now, for each $e, e' \notin E$, we randomly swap an their operations $(e, o)$ and $(e', o')$ that are within a distance of $w$ while maintaining consistency.

## 7.2 Experiment Results

**Experiment Set 1.** We first consider the effect of the prediction error $\eta$ on the function value. For ease of exposition, we overload the notation and report the fractional prediction error, i.e. prediction error divided by the length of the stream $n$. From the first column of Figure 1, we observe that our algorithm outperforms OFFLINEGREEDY in terms of function value when the prediction error is reasonably large, and always achieves a similar function value as DYNAMIC. Since DYNAMIC does not use the predictions, its performance remains constant as a function of the prediction errors. The performance of OFFLINEGREEDY deteriorates quickly as a function of $\eta$ as it completely relies on the predictions. Note that the function value achieved by OFFLINEGREEDY is not zero even in the case of large prediction error. This is because the error is not adversarial and some elements from its offline solution remain active during the streaming phase.

We also consider the effect of $\eta$ on the number of oracle calls. Figure 1 shows that the number of oracle calls of our algorithm is much better than DYNAMIC in the case of low prediction error, and is also not much worse in the case of large prediction error. This shows that our algorithm has consistency in the case of low $\eta$, but also robustness in the case of large $\eta$. The number of oracle calls of OFFLINEGREEDY is very small as it completely relies on the prediction.

**Experiment Set 2.** We also consider the effect of cardinality parameter $k$ on the function value and number of oracle calls. It can be observed from the second column of Figure 1 that the function value increases for all algorithms as a function of $k$. Moreover, the rate of increase for the number of oracle calls made by our algorithm is similar to the rate of increase of DYNAMIC. **Experiment Set 3.** We consider the effect of the window size parameter $w$ on the function value and number of oracle calls. Figure 1 shows that the window size has almost no impact on the function value of our algorithm. The number of oracle calls for our algorithm grows as a function of the window size but this growth is very small.

## 8 Limitations

To obtain an asymptotic improvement over the best-known amortized query complexity per update, the prediction error $\eta$ needs to be subpolynomial in the length of the stream $n$, which is relatively small. However, our experimental results show that in practice the number of queries performed by our algorithm outperforms the number of queries of existing algorithm without predictions even when the prediction error is relatively large. Another limitation is the assumption that the algorithm is given all predictions at time $t = 0$. An intriguing open question is whether an improvement in update time can also be obtained in the predicted-deletion model where there is no preprocessing and instead a predicted deletion time for each element is given at the time when it is inserted.

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

# A  Missing proofs from Section 3

**Lemma 8.** *Assume that the predicted stream has, with a time error tolerance $w$, a prediction error at most $\eta$. Then, at any time step $t$, $|\hat{V}_t \setminus V_t| \leq \eta + 2w$.*

*Proof.* Let $E = \{a \in V : |\hat{t}_a^+ - t_a^+| > w \text{ or } |\hat{t}_a^- - t_a^-| > w\}$. Hence, for all $a \notin E$, we have $|\hat{t}_a^+ - t_a^+| \leq w$ and $|\hat{t}_a^- - t_a^-| \leq w$. By Definition 2, we have that $|E| = \eta$. We first note that any $a \in \hat{V}_t \setminus E$ such that $\hat{t}_a^+ \leq t - w$ and $\hat{t}_a^- > t + w$ also belongs to $V_t$. This is because $a \notin E$ implies that $t_a^+ \leq t$ and $t_a^- > t$. Hence, the only elements that are present in $\hat{V}_t \setminus E$ but are absent from $V_t$ are such that $\hat{t}_a^+ > t - w$ or $\hat{t}_a^- \leq t + w$. Since, the only elements in $\hat{V}_t$ are such that $\hat{t}_a^+ \leq t + w$ and $\hat{t}_a^- \geq t - w$, this implies that there can be at most $2w$ such elements. Combining with the fact that $|E| = \eta$, we get $|\hat{V}_t \setminus V_t| \leq |E| + |(\hat{V}_t \setminus E) \setminus V_t| \leq \eta + 2w$. $\qquad\square$

**Lemma 1.** *At every time step $t$, $\mathbf{E}[f(S^1)] \geq \alpha_1 \cdot \mathit{OPT}(V_t^1)$ and $\mathbf{E}[f(S^2)] \geq \alpha_2 \cdot \mathit{OPT}(V_t^2)$.*

*Proof.* We first show that at every time step $t$, $f(S^1) \geq \alpha_1 \cdot \text{OPT}(V_t^1)$. Fix some time step $t$. First, note that

$$P_t \setminus (\hat{V}_t \setminus V_t^1) = (P_t \setminus \hat{V}_t) \cup (P_t \cap V_t^1) = P_t \cap V_t^1$$

where the second equality is since $P_t \subseteq \hat{V}_t$. In addition, we have that $|\hat{V}_t \setminus V_t^1| = |\hat{V}_t \setminus V_t| \leq \eta + 2w = d$, where the inequality is by Lemma 8. With $D = \hat{V}_t \setminus V_t^1$ such that $|D| \leq d$, $R = P_t$, and $N = \hat{V}_t$, we thus have by Definition 3 that the output $S^1$ of $\text{ROBUST2}(f, P_t, \hat{V}_t \setminus V_t^1, k) = \text{ROBUST2}(f, R, D, k)$ is such that $S^1 \subseteq P_t \cap V_t^1$ and

$$\mathbf{E}[f(S^1)] \geq \alpha_1 \cdot \max_{\substack{S \subseteq \hat{V}_t \setminus D: \\ |S| \leq k}} f(S) = \alpha_1 \cdot \max_{\substack{S \subseteq V_t^1: \\ |S| \leq k}} f(S) = \alpha_1 \cdot \text{OPT}(V_t^1).$$

Next, we show that at every time step $t$, $f(S^2) \geq \alpha_2 \cdot \text{OPT}(V_t^2)$. Observe that the algorithm runs the dynamic streaming algorithm $\text{DYNAMIC}$ over the sequence of ground sets $V_0^2, \ldots, V_n^2$. By Definition 1, the solutions returned by $\text{DYNAMIC}$ at each time step achieve an $\alpha_2$ approximation. Since these solutions are $S_1^2, \ldots, S_n^2$, we get that for every time step $t$, $f(S_t^2) \geq \alpha_2 \max_{S \subseteq V_t^2 : |S| \leq k} f(S)$. $\qquad\square$

**Lemma 9.** *For any monotone submodular function $f : 2^{N_1 \cup N_2} \to \mathbb{R}$, if $S_1$ and $S_2$ are such that $f(S_1) \geq \alpha_1 \cdot \max_{S \subseteq N_1 : |S| \leq k} f(S)$ and $f(S_2) \geq \alpha_2 \cdot \max_{S \subseteq N_2 : |S| \leq k} f(S)$, then $\max\{f(S_1), f(S_2)\} \geq \frac{1}{2} \cdot \min\{\alpha_1, \alpha_2\} \cdot \max_{S \subseteq N_1 \cup N_2 : |S| \leq k} f(S)$.*

*Proof.* Let $O = \{o_1, \ldots, o_k\} = \text{argmax}_{S \subseteq N_1 \cup N_2 : |S| \leq k} f(S)$ be an arbitrary ordering of the optimal elements. We have that

$$\max_{S \subseteq N_1 \cup N_2 : |S| \leq k} f(S) = f(O)$$

$$= \sum_{i=1}^{k} f_{\{o_1, \cdots, o_{i-1}\}}(o_i)$$

$$\leq \sum_{i=1}^{k} \mathbb{1}[o_i \in N_1] f_{\{o_1, \cdots, o_{i-1}\} \cap N_1}(o_i) + \sum_{i=1}^{k} \mathbb{1}[o_i \in N_2] f_{\{o_1, \cdots, o_{i-1}\} \cap N_2}(o_i)$$

$$= f(O \cap N_1) + f(O \cap N_2)$$

$$\leq \max_{S \subseteq N_1 : |S| \leq k} f(S) + \max_{S \subseteq N_2 : |S| \leq k} f(S)$$

$$\leq f(S_1)/\alpha_1 + f(S_2)/\alpha_2$$

$$\leq 2 \max\{f(S_1), f(S_2)\}/\min\{\alpha_1, \alpha_2\}$$

where the first inequality is by submodularity. $\qquad\square$

By combining Lemma 1 and Lemma 9, we obtain the main lemma for the approximation guarantee.

**Lemma 10.** *At every time step $t$, $\max\{f(S^1), f(S^2)\} \geq \frac{1}{2} \cdot \min\{\alpha_2, \alpha_1\} \cdot \mathit{OPT}(V_t)$.*

**Lemma 2.** WARMUP-UPDATESOL *makes at most $2\eta$ calls to* DYNAMICINS *on $A$ over the stream.*

*Proof.* The number of calls to DYNAMICINS is $|\{a : \exists t \text{ s.t. } a \in V_t^2 \setminus V_{t-1}^2\}|$. Since $V_t^2 \setminus V_{t-1}^2 = (V_t \setminus \hat{V}_t) \setminus (V_{t-1} \setminus \hat{V}_{t-1}^1)$,

$$|\{a : \exists t \text{ s.t. } a \in V_t^2 \setminus V_{t-1}^2\}| \leq |\{a : \exists t \text{ s.t. } a \in (V_t \setminus V_{t-1}) \setminus \hat{V}_t\}|$$
$$+ |\{a : \exists t \text{ s.t. } a \in (\hat{V}_{t-1} \setminus \hat{V}_t) \cap V_t\}|.$$

Next, we have

$$|\{a : \exists t \text{ s.t. } a \in (V_t \setminus V_{t-1}) \setminus \hat{V}_t\}| = |\{a : a \notin \hat{V}_{t_a^+}\}|$$
$$= |\{a : \hat{t}_a^+ > t_a^+ + w \text{ or } \hat{t}_a^- < t_a^+ - w\}|$$
$$\leq |\{a : \hat{t}_a^+ > t_a^+ + w \text{ or } \hat{t}_a^- < t_a^- - w\}|$$
$$\leq |\{a : |\hat{t}_a^+ - t_a^+| > w \text{ or } |\hat{t}_a^- - t_a^-| > w\}|$$
$$\leq \eta$$

Similarly,

$$|\{a : \exists t \text{ s.t. } a \in (\hat{V}_{t-1} \setminus \hat{V}_t) \cap V_t\}| = |\{a : a \in V_{\hat{t}_a^- + w}\}|$$
$$= |\{a : t_a^- \geq \hat{t}_a^- + w\}|$$
$$\leq |\{a : |\hat{t}_a^+ - t_a^+| > w \text{ or } |\hat{t}_a^- - t_a^-| > w\}|$$
$$\leq \eta$$

Combining the above inequalities, we get that the number of calls to DYNAMICINS is $|\{a : \exists t \text{ s.t. } a \in V_t^2 \setminus V_{t-1}^2\}| \leq 2\eta$. $\square$

**Theorem 1.** *For monotone submodular maximization under a cardinality constraint $k$, Algorithm 1 with the* WARMUP-PRECOMPUTATIONS *and* WARMUP-UPDATESOL *subroutines achieves, for any tolerance $w$ and constant $\epsilon > 0$, an amortized expected query complexity per update during the streaming phase of $\tilde{O}(\eta + w + k)$, an approximation of $1/4 - \epsilon$ in expectation, and a query complexity of $\tilde{O}(n^2 k)$ during the precomputation phase.*

*Proof.* We choose DYNAMIC to be the algorithm from [20] and ROBUST to be the algorithm from [16]. By Lemma 10, and since we have $\alpha_1 = \alpha_2 = 1/2 - \epsilon$, the approximation is $\frac{1}{2} \cdot \min\{\alpha_2, \alpha_1\} = \frac{1}{2} \cdot \min\{\frac{1}{2} - \epsilon, \frac{1}{2} - \epsilon\} \geq \frac{1}{4} - \epsilon$.

For the query complexity during the streaming phase, by Lemma 2, the total number of calls to DYNAMICINS on $A$ is $O(\eta)$. The total number of calls to DYNAMICDEL on $A$ is also $O(\eta)$ since an element can only be deleted if it has been inserted. Thus, the total number of insertions and deletions handled by the dynamic streaming algorithm DYNAMIC is $O(\eta)$. Since the amortized expected query complexity per update of DYNAMIC is $O(\text{poly}(\log n, \log k))$, we get that the amortized expected number of queries performed when calling DYNAMICINS, DYNAMICDEL, and DYNAMICSOL on $A$ is $O(\eta \cdot \text{poly}(\log \eta, \log k)/n)$. For the number of queries due to ROBUST2, at every time step $t$, the algorithm calls ROBUST2$(f, P_t, (\hat{V}_t \setminus V^1), k)$ with maximum number of deletions $d = \eta + 2w$, which causes at most $\tilde{O}(\eta + w + k)$ queries at each time step $t$ since the query complexity of ROBUST2 is $\tilde{O}(d + k)$. The total amortized expected query complexity per update during the streaming phase is thus $\tilde{O}(\eta + w + k)$. For the query complexity during the precomputation phase, the query complexity of ROBUST1 is $\tilde{O}(|V|k)$ and there are $n$ calls to ROBUST1, so the query complexity of that phase is $\tilde{O}(n^2 k)$. $\square$

# B   Missing proof from Section 4

**Lemma 3.** *The* DYNAMIC *algorithm of Lattanzi et al. [20][6] is threshold-based.*

*Proof.* The first condition that $f(\text{SOL}_t) \geq \frac{\gamma}{2k}|\text{SOL}_t|$ follows directly from the fact that each element $e$ that is added to $\text{SOL}_t$ has a marginal contribution of at least $\gamma/2k$ to the partial solution. Some elements could have been deleted from the partial solution since the time $e$ was added, but this can only increase the contribution of $e$ due to submodularity.

We will now show that if $|\text{SOL}_t| < (1-\epsilon)k$ then for any $S \subseteq V(A_t)$ we have $f_{\text{SOL}_t}(S) < \frac{|S|\gamma}{2k} + \epsilon\gamma$. We will use the set $X$ defined in the proof of Theorem 5.1 in [20]. The proof of Theorem 5.1 in [20] shows that $|\text{SOL}_t| \geq (1-\epsilon)|X|$. Using this, the condition that $|\text{SOL}_t| < (1-\epsilon)k$ implies that $|X| < k$. Hence, we fall in the case 2 of the proof of Theorem 5.1 which shows that $f(e|X) \leq \gamma/2k$ for all $e \in V_t \setminus \text{SOL}_t$. We then have that

$$
\begin{aligned}
f_{\text{SOL}_t}(S) = f(S \cup \text{SOL}_t) - f(\text{SOL}_t) &\leq f(S \cup X) - f(\text{SOL}_t) \\
&= f(S \cup X) - f(X) + f(X) - f(\text{SOL}_t) \\
&\leq f(X \cup S) - f(X) + f(X) - (1-\epsilon)f(X) \\
&\leq \sum_{e \in S} f_X(e) + \epsilon f(X) \\
&\leq |S| \cdot \frac{\gamma}{2k} + \epsilon \cdot \frac{1+\epsilon}{1-\epsilon} \cdot \gamma,
\end{aligned}
$$

where the inequality $f(X) \leq \frac{1+\epsilon}{1-\epsilon}\gamma$ follows because $f(X) \leq f(\text{SOL}_t)/(1-\epsilon) \leq \frac{1+\epsilon}{1-\epsilon} \cdot \gamma$.   $\square$

**Lemma 4.** *Consider the data structure $(B, A, \eta_{old})$ returned by* UPDATESOLMAIN *at time $t$. Let $t'$ be the time at which $A$ was initialized, $(Q_{t'}, R_{t'})$ and $\gamma_{t'}$ be the precomputations and guess for $\text{OPT}_{t'}$ inputs to* UPDATESOLMAIN *at time $t'$. If $(Q_{t'}, R_{t'})$ are $(d = 2(\eta_{old} + 2w), \epsilon, \gamma_{t'})$-strongly robust, $\gamma_{t'}$ is such that $\gamma_{t'} \leq \text{OPT}_t \leq (1+\epsilon)\gamma_{t'}$, and* DYNAMIC *is a threshold-based dynamic algorithm, then the set $S$ returned by* UPDATESOLMAIN *is such that $\mathbf{E}[f(S)] \geq \frac{1-5\epsilon}{2}\gamma_{t'}$.*

To prove Lemma 4, we first bound the value of $Q_{t'} \cap V_t$.

**Lemma 11.** *Consider the data structure $(B, A, \eta_{old})$ returned by* UPDATESOLMAIN *at time $t$. Let $t'$ be the time at which $A$ was initialized, $(Q_{t'}, R_{t'})$ and $\gamma_{t'}$ be the precomputations and guess for $\text{OPT}_{t'}$ inputs to* UPDATESOLMAIN *at time $t'$. If precomputations $(Q_{t'}, R_{t'})$ are $(d = 2(\eta_{old} + 2w), \epsilon, \gamma_{t'})$-strongly robust, then we have*

$$
\mathbf{E}[f(Q_{t'} \cap V_t)] \geq (1-\epsilon)f(Q_{t'}) \geq (1-\epsilon)|Q_{t'}|\gamma_{t'}/(2k).
$$

*Proof.* We first show that $|Q_{t'} \setminus V_t| \leq d$. We know that $Q_{t'} \subseteq \hat{V}_{t'}$. Firstly, we have $|\hat{V}_{t'} \setminus V_{t'}| \leq \eta_t + 2w \leq \eta_{old} + 2w$ due to the Lemma 8. Next, note that the number of insertions and deletions $\text{Ops}^\star(A)$ to $A$ between time $t'$ and $t$ is at most $\frac{\eta_{old}}{2} + w$, otherwise $A$ would have been reinitialized due to the condition of UPDATESOLMAIN to start a new phase. This implies that $|V_{t'} \setminus V_t| \leq \eta_{old} + 2w$. Hence, we have that

$$
|Q_{t'} \setminus V_t| \leq |\hat{V}_{t'} \setminus V_t| \leq |\hat{V}_{t'} \setminus V_{t'}| + |V_{t'} \setminus V_t| \leq 2(\eta_{old} + 2w) = d.
$$

We conclude that $\mathbf{E}[f(Q_{t'} \cap V_t)] \geq (1-\epsilon)f(Q_{t'}) \geq (1-\epsilon)|Q_{t'}|\frac{\gamma_{t'}}{2k}$, where the first inequality is by the robustness property of strongly-robust precomputations and the second by their value property.   $\square$

*Proof of Lemma 4.* There are three cases.

1. $|Q_{t'}| \geq k$. We have that $\mathbf{E}[f(S)] \geq \mathbf{E}[f(Q_{t'} \cap V_t)] \geq (1-\epsilon)|Q_{t'}|\frac{\gamma_{t'}}{2k} \geq (1-\epsilon)\frac{\gamma_{t'}}{2}$, where the first inequality is by monotonicity and since $S \supseteq B \cap V_t = Q_{t'} \cap V_t$, the second by Lemma 11, and the last by the condition for this first case.

---

[6]Note that the initially published algorithm in [20] had an issue with correctness, we refer to the revised version.

2. $|\text{DYNAMICSOL}(A)| \geq (1 - \epsilon)(k - |Q_{t'}|)$. Let $\text{SOL}_t = \text{DYNAMICSOL}(A)$ and recall that $A$ was initialized with DYNAMICINIT over function $f_{Q_{t'}}$ with cardinality constraint $k - |B|$ and threshold parameter $\gamma_{t'}(k - |B|)/k$. We get

$$
\begin{aligned}
\mathbf{E}[f(S)] &= \mathbf{E}[f(Q_{t'} \cap V_t) + f_{Q_{t'} \cap V_t}(\text{SOL}_t)] && \text{definition of } S \\
&\geq \mathbf{E}[f(Q_{t'} \cap V_t) + f_{Q_{t'}}(\text{SOL}_t)] && \text{submodularity} \\
&\geq (1 - \epsilon)|Q_{t'}|\frac{\gamma_{t'}}{2k} + \mathbf{E}[f_{Q_{t'}}(\text{SOL}_t)] && \text{Lemma 11} \\
&\geq (1 - \epsilon)|Q_{t'}|\frac{\gamma_{t'}}{2k} + \frac{\gamma_{t'}(k - |B|)/k}{2(k - |B|)}|\text{SOL}_t| && \text{Definition 5} \\
&\geq (1 - \epsilon)|Q_{t'}|\frac{\gamma_{t'}}{2k} + \frac{\gamma_{t'}}{2k}(1 - \epsilon)(k - |Q_{t'}|) && \text{case assumption} \\
&\geq \frac{(1 - \epsilon)\gamma_{t'}}{2}.
\end{aligned}
$$

3. $|Q_{t'}| < k$ and $|\text{SOL}_t| < (1 - \epsilon)(k - |Q_{t'}|)$: Recall that $R_{t'} \subseteq \hat{V}_{t'}$. Also, let $\bar{R}_t = (V_t \cap R_{t'}) \cup (V_t \setminus \hat{V}_{t'})$. In this case, we have that

$$
\begin{aligned}
f(O_t) \leq_{(a)}\ & \mathbf{E}[f(O_t \cup \text{SOL}_t \cup Q_{t'})] \\
=\ & \mathbf{E}[f(\text{SOL}_t \cup Q_{t'})] + \mathbf{E}[f_{\text{SOL}_t \cup Q_{t'}}(O_t \setminus \bar{R}_t)] + \mathbf{E}[f_{\text{SOL}_t \cup Q_{t'} \cup (O_t \setminus \bar{R}_t)}(O_t \cap \bar{R}_t)] \\
\leq_{(b)}\ & \mathbf{E}[f(\text{SOL}_t \cup Q_{t'})] + \mathbf{E}[f_{Q_{t'}}(O_t \setminus \bar{R}_t)] + \mathbf{E}[f_{\text{SOL}_t}(O_t \cap \bar{R}_t)] \\
\leq_{(c)}\ & \mathbf{E}[f(\text{SOL}_t \cup Q_{t'})] + |O_t \setminus \bar{R}_t| \cdot \frac{\gamma_{t'}}{2k} + \epsilon\gamma_{t'} + \mathbf{E}[f_{\text{SOL}_t}(O_t \cap \bar{R}_t)] \\
\leq_{(d)}\ & \mathbf{E}[f(\text{SOL}_t \cup Q_{t'})] + |O_t \setminus \bar{R}_t| \cdot \frac{\gamma_{t'}}{2k} + \epsilon\gamma_{t'} + |O_t \cap \bar{R}_t| \cdot \frac{\gamma_{t'}}{2k} + \epsilon\gamma_{t'} \\
\leq_{(e)}\ & \mathbf{E}[f(\text{SOL}_t \cup Q_{t'})] + (1 + 4\epsilon)\frac{\gamma_{t'}}{2}.
\end{aligned}
$$

where $(a)$ is by monotonicity, $(b)$ is by submodularity, $(c)$ is since $(Q_{t'}, R_{t'})$ are $(d = 2(\eta_{\text{old}} + 2w), \epsilon, \gamma_{t'})$-strongly robust (value property) and since we have that $O_t \setminus \bar{R}_t = O_t \setminus ((V_t \cap R_{t'}) \cup (V_t \setminus \hat{V}_{t'})) \subseteq \hat{V}_{t'} \setminus R_{t'}$ and $Q_{t'} < k$, $(d)$ is since DYNAMIC is threshold-based, $|\text{SOL}_t| < (1 - \epsilon)(k - |Q_{t'}|)$, and $\bar{R}_t \subseteq V(A)$, and $(e)$ is since $|O_t| \leq k$. The above series of inequalities implies that

$$
\mathbf{E}[f(\text{SOL}_t \cup Q_{t'})] \geq f(O_t) - \frac{(1 + 4\epsilon)\gamma_{t'}}{2} = \frac{(1 - 4\epsilon)\gamma_{t'}}{2}.
$$

We conclude that

$$
\begin{aligned}
\mathbf{E}[f(S)] &= \mathbf{E}[f((Q_{t'} \cup \text{SOL}_t) \cap V_t)] && \text{definition of } S \\
&= \mathbf{E}[f((Q_{t'} \cap V_t) \cup \text{SOL}_t)] && \text{SOL}_t \subseteq V_t \\
&\geq \mathbf{E}[f(Q_{t'} \cap V_t)] + \mathbf{E}[f_{Q_{t'}}(\text{SOL}_t)] && \text{submodularity} \\
&\geq (1 - \epsilon)f(Q_{t'}) + \mathbf{E}[f_{Q_{t'}}(\text{SOL}_t)] && \text{Lemma 11} \\
&\geq (1 - \epsilon) \cdot \mathbf{E}[f(\text{SOL}_t \cup Q_{t'})] \\
&\geq \frac{(1 - 5\epsilon)\gamma_{t'}}{2} && \square
\end{aligned}
$$

**Lemma 5.** *Consider the data structure $(B, A, \eta_{\text{old}})$ returned by UPDATESOLMAIN at time $t$. Let $t'$ be the time at which $A$ was initialized and $(Q_{t'}, R_{t'})$ be the precomputations input to UPDATESOLMAIN at time $t'$. If precomputations $(Q_{t'}, R_{t'})$ are $(d = 2(\eta_{\text{old}} + 2w), \epsilon, \gamma)$ strongly-robust, then the total number of queries performed by UPDATESOL during the $t - t'$ time steps between time $t'$ and time $t$ is $O(u((\eta_{\text{old}} + w + k)\log k, k) \cdot (\eta_{\text{old}} + w + k)\log k)$. Additionally, the number of queries between time 1 and $t$ is upper bounded by $O(u(t, k) \cdot t)$.*

*Proof.* The only queries made by UPDATESOL are due to calls to DYNAMIC. Hence, we calculate the total number of operations $\text{Ops}(A)$ between time $t'$ and $t$. The number of insertions at time $t'$ is

$$
\begin{aligned}
|R_{t'} \cap V_{t'}^1| + |V_{t'}^2| &=_{(1)} O((\eta_{\text{old}} + w + k)\log k) + |V_{t'}^2| \\
&\leq_{(2)} O((\eta_{\text{old}} + w + k)\log k) + \eta_{\text{old}} \\
&= O((\eta_{\text{old}} + w + k)\log k)),
\end{aligned}
$$

where (1) is by the size property of the $(d = 2(\eta_{\text{old}} + 2w), \epsilon, \gamma)$ strongly-robust precomputations and (2) is since $|V_{t'}^2| \leq \eta_{t'} \leq \eta'_{t'} = \eta_{\text{old}}$.

Moreover, the total number of operations $\text{Ops}^*(A)$ between time $t' + 1$ and $t$ is at most $\eta_{\text{old}}/2 + w = d/4$, otherwise $A$ would have been reinitialized due to the condition of UPDATESOLMAIN to start a new phase. Hence, the total query complexity between time $t'$ and time $t$ is at most $u(O((\eta_{\text{old}} + w + k) \log k), k - |Q_{t'}|) \cdot O((\eta_{\text{old}} + w + k) \log k) = O(u((\eta_{\text{old}} + w + k) \log k, k) \cdot (\eta_{\text{old}} + w + k) \log k)$.

In the case $t' = 1$, the number of operations at time $t'$ is 1 since $|V_{t'}| = 1$, and the number of operations from time 2 to $t$ is at most $t - 1$. This gives the bound of $O(u(t, k) \cdot t)$ when $t' = 1$. $\qquad\square$

## C  Missing proofs from Section 5

**Lemma 6.** *Let* DYNAMIC$(\gamma, \epsilon)$ *be the dynamic submodular maximization algorithm of [20] and $A$ be the data structure it maintains. There is a* ROBUST1FROMDYNAMIC *algorithm such that, given as input a deletion size parameter $d$, and the data structure $A$ at time $t$ with $\gamma \leq \text{OPT}_t \leq (1 + \epsilon)\gamma$, it outputs sets $(Q, R)$ that are $(d, \epsilon, \gamma)$-strongly robust with respect to the ground set $V_t$. Moreover, this algorithm does not perform any oracle queries.*

*Proof.* We will first set up some notation. The algorithm of [20] creates several buckets $\{A_{i,\ell}\}_{i \in [R], \ell \in [T]}$ where $R = O(\log k / \epsilon)$ is the number of thresholds and $T = O(\log n)$ is the number of levels. A solution is constructed using these buckets by iteratively selecting $S_{i,\ell}$ from $A_{i,\ell}$ starting from the smallest $(i, \ell) = (0, 0)$ to the largest $(i, \ell) = (R, T)$. The buffer at level $\ell$ is denoted by $B_\ell$. Each set $S_{i,\ell}$ is constructed by iteratively selecting elements uniformly at random (without replacement) from the corresponding bucket. More precisely, given indices $i, \ell$, the algorithm adds elements $e$ to the solution $S_{i,\ell}$ one by one in line 7 of Algorithm 6 in [20] only if $f(e|S) \geq \tau_i \geq \gamma/2k$.

We will now show how to construct $(Q, R)$ from the DYNAMIC data structure $A$. Note that we will extract $(Q, R)$ by observing the evolution of the data-structure $A$. Hence, we do not need to make any additional oracle queries in order to extract $(Q, R)$.

The following algorithm gives a procedure for extracting $(Q, R)$ by observing the actions of the peeling algorithm (Algorithm 6) in [20].

---

**Algorithm 5** ROBUST1FROMDYNAMIC

**Input:** function $f : 2^V \to \mathbb{R}$, dynamic data-structure $A$, constraint $k$, deletion parameter $d$, parameter $\epsilon$
1: $Q \leftarrow \emptyset$
2: $R \leftarrow \emptyset$
3: **for** $\ell \in [T]$ **do**
4:     **for** $i \in [R]$ **do**
5:         **if** $n/2^\ell > \frac{d}{\epsilon} + k$ **then**
6:             $Q \leftarrow Q \cup S_{i,\ell}$
7:         **else**
8:             $R \leftarrow R \cup A_{i,\ell} \cup B_\ell$
9: **return** $Q, R$

---

We will now show that the conditions required in Definition 4 by the above algorithm.

- **Size.** The fact that $|Q| \leq k$ follows using the $Q \subseteq \cup_{i,\ell} S_{i,\ell} = S$ where $S$ is the solution output by DYNAMIC with $|S| \leq k$.

  We will now show that the size of $R = O(\frac{\log k}{\epsilon} \cdot (\frac{d}{\epsilon} + k))$. Let $\bar{\ell} \in [T]$ be the largest value such that the corresponding bucket size $n/2^{\bar{\ell}}$ is at least $d/\epsilon + k$. Recall that

$$R = \cup_{\ell < \bar{\ell}} B_\ell \cup \left( \cup_{i \in [R]} A_{i,\ell} \right).$$

We first have that

$$| \cup_{\ell < \bar{\ell}} B_\ell| \leq \sum_{\ell < \bar{\ell}} n/2^\ell \leq 2n/2^{\bar{\ell}-1} = O(\frac{d}{\epsilon} + k) \,.$$

Secondly,

$$| \cup_{\ell < \bar{\ell}} \cup_{i \in [R]} A_{i,\ell}| \leq \sum_{\ell < \bar{\ell}} \sum_{i \in [R]} n/2^\ell \leq \frac{\log k}{\epsilon} \cdot 2n/2^{\bar{\ell}-1} = O\left(\frac{\log k}{\epsilon} \cdot \left(\frac{d}{\epsilon} + k\right)\right) \,.$$

Combining the above inequalities gives us the desired bound on the size of $R$.

- **Value.** The fact that $f(Q) \geq |Q|\gamma/2k$ follows directly using the property of the dynamic algorithm of [20] an element $e$ is added to $S_{i,\ell}$ during a call to Bucket-Construct$(i, \ell)$ only if $f_{S'}(e) \geq \tau_i$ where $\tau_i \geq \gamma/2k$ and $S'$ is the partial solution. It is possible that some of the elements of the partial solution have been deleted since the last time Bucket-Construct was called on index $(i, \ell)$, but the marginal contribution of $e$ can only increase with deletions due to submodularity.

  We now show that, if $|Q| < k$, then for any for any set $S \subseteq V \setminus R$ we have $f_Q(S) < |S|\gamma/(2k) + \epsilon\gamma$. Let $Q'$ denote the set of elements in $Q$ at the time when Level-Construct$(\bar{\ell})$ was called last. We have

$$\begin{aligned} f_Q(S) = f(Q \cup S) - f(Q) &\leq f(Q' \cup S) - f(Q) \\ &\leq f(Q' \cup S) - f(Q') + f(Q') - f(Q) \\ &\leq f(Q' \cup S) - f(Q') + f(Q') - (1-\epsilon)f(Q') \\ &\leq \sum_{e \in S} f_{Q'}(e) + \epsilon f(Q') \\ &\leq |S| \cdot \frac{\gamma}{2k} + \epsilon \cdot \frac{1+\epsilon}{1-\epsilon} \cdot \gamma \,, \end{aligned}$$

  where the inequality $f(Q') \leq \frac{1+\epsilon}{1-\epsilon}\gamma$ follows because $f(Q') \leq f(Q)/(1-\epsilon) \leq \frac{1+\epsilon}{1-\epsilon} \cdot \gamma$, and the inequality $f_{Q'}(e) \leq \gamma/2k$ follows by considering two cases: (1) $e$ was inserted before the latest time Level-Construct$(\bar{\ell})$ was called in which case $e$ cannot have high marginal contribution to $Q'$ as otherwise it would have been inserted in $Q'$, (2) $e$ was inserted after the latest time Level-Construct$(\bar{\ell})$ was called in which case either $e$ has low marginal contribution to $Q'$ or it would be included in the set $R$.

- **Robustness.** For any set $D \subseteq V$ such that $|D| \leq d$, $\mathbf{E}_Q[f(Q \setminus D)] \geq (1-\epsilon)f(Q)$. We have that

$$f(Q) = \sum_{\ell \geq \bar{\ell}} \sum_{i \in [R]} \sum_{j \in [|S_{i,\ell}|]} f_{S_{\ell,i,j}^{partial}}(e_{\ell,i,j}) \,.$$

  where $e_{\ell,i,j}$ is the $j$-th element that was added to $S_{i,\ell}$ and $S_{\ell,i,j}^{partial}$ is the partial solution before the $j$-th element was added. We have that

$$\sum_{\ell \geq \bar{\ell}} \sum_{i \in [R]} |S_{i,\ell}|\tau_i \geq f(Q) \leq (1+\epsilon) \sum_{\ell \geq \bar{\ell}} \sum_{i \in [R]} |S_{i,\ell}|\tau_i$$

Since, each element in $S_{i,\ell}$ is selected from a set of size at least $d/\epsilon$, it can only be deleted with probability at most $\epsilon$. We have that

$$
\begin{aligned}
\mathbf{E}[f(Q \setminus D)] &= \sum_{\ell \geq \bar{\ell}} \sum_{i \in [R]} \sum_{j \in [|S_{i,\ell}|]} \mathbf{E}[\mathbb{1}[e_{\ell,i,j} \notin D] \cdot f_{A_{\ell,i,j}^{partial}}(e_{\ell,i,j})] \\
&\geq \sum_{\ell \geq \bar{\ell}} \sum_{i \in [R]} \sum_{j \in [|S_{i,\ell}|]} \mathbf{E}[\mathbb{1}[e_{\ell,i,j} \notin D] \cdot f_{S_{\ell,i,j}^{partial}}(e_{\ell,i,j})] \\
&\geq \sum_{\ell \geq \bar{\ell}} \sum_{i \in [R]} \sum_{j \in [|S_{i,\ell}|]} \Pr(e_{\ell,i,j} \notin D) \cdot \tau_i \\
&\geq (1 + \epsilon) \sum_{\ell \geq \bar{\ell}} \sum_{i \in [R]} |S_{i,\ell}| \tau_i \\
&\geq \frac{1+\epsilon}{1-\epsilon} f(Q). \quad \square
\end{aligned}
$$

**Lemma 7.** *The total query complexity of the* PRECOMPUTATIONSMAIN *algorithm is* $n \cdot u(n, k)$, *where* $u(n, k)$ *is the amortized query complexity of calls to* DYNAMIC.

*Proof.* It is easy to observe that the algorithm makes at most $n$ calls to DYNAMICINS or DYNAMICDEL since $\sum_t |\hat{V}_t \setminus \hat{V}_{t-1}| + |\hat{V}_{t-1} \setminus \hat{V}_t| \leq n$. Hence, the total query complexity due to calls to DYNAMIC is $n \cdot u(n, k)$. Moreover, the calls to ROBUST1FROMDYNAMIC do not incur any additional queries due to Lemma 6. Hence, the total number of queries performed in the precomputation phase is given by $n \cdot u(n, k)$. $\square$

# D  Missing proofs from Section 6

The UPDATESOLMAIN and PRECOMPUTATIONSMAIN subroutines use guesses $\gamma_t$ and $h$ for the optimal value $\mathrm{OPT}_t$ at time $t$ and the total prediction error $\eta$. In this section, we describe the full UPDATESOLFULL and PRECOMPUTATIONSFULL subroutines that call the previously defined subroutines over different guesses $\gamma_t$ and $h$. The parameters of these calls must be carefully designed to bound the streaming amortized query complexity per update and precomputations query complexity.

## D.1  The full PRECOMPUTATIONS subroutine

We define $H = \{n/2^i : i \in \{\log_2(\max\{\frac{n}{k-2w}, 1\}), \cdots, \log_2(n) - 1, \log_2 n\}\}$ to be a set of guesses for the prediction error $\eta$. Since PRECOMPUTATIONSMAIN requires a guess $h$ for $\eta$, PRECOMPUTATIONSFULL calls PRECOMPUTATIONSMAIN over all guesses $h \in H$. We ensure that the minimum guess $h$ is such that $h + 2w$ is at least $k$. The challenge with the guess $\gamma$ of OPT needed for PRECOMPUTATIONSMAIN is that OPT can have arbitrarily large changes between two time steps. Thus, with $\hat{V}_1, \ldots, \hat{V}_n \subseteq V$, we define the collection of guesses for OPT to be $\Gamma = \{(1 + \epsilon)^0 \min_{a \in V} f(a), (1 + \epsilon)^1 \min_{a \in V} f(a), \ldots, f(V)\}$, which can be an arbitrarily large set.

Instead of having a bounded number of guesses $\gamma$, we consider a subset $\hat{V}_t(\gamma) \subseteq \hat{V}_t$ of the predicted elements at time $t$ for each guess $\gamma \in \Gamma$ such that for each element $a$ is, there is a bounded number of guesses $\gamma \in \Gamma$ such that $a \in \hat{V}_t(\gamma)$. More precisely, for any set $T$, we define $T(\gamma) := \{a \in T : \frac{\epsilon\gamma}{k} \leq f(a) \leq 2\gamma\}$ and $\Gamma(a) = \{\gamma \in \Gamma : f(a) \leq \gamma \leq \epsilon^{-1}kf(a)\}$. The PRECOMPUTATIONSFULL subroutine outputs, for every time step $t$, strongly-robust sets $(Q_t^{\gamma,h}, R_t^{\gamma,h})$, for all guesses $\gamma \in \Gamma$ and $h \in H$. If $\hat{V}_t(\gamma) = \emptyset$, we assume that $Q_t^{\gamma,h} = \emptyset$ and $R_t^{\gamma,h} = \emptyset$.

---

**Algorithm 6** PRECOMPUTATIONSFULL

---

**Input:** function $f$, constraint $k$, predicted active elements $\hat{V}_1, \ldots, \hat{V}_n \subseteq V$, time error tolerance $w$

1: **for** $\gamma \in \Gamma$ such that $|\hat{V}_t(\gamma)| > 0$ for some $t \in [n]$ **do**
2:     **for** $h \in H$ **do**
3:         $\{(Q_t^{\gamma,h}, R_t^{\gamma,h})\}_{t=1}^n \leftarrow$ PRECOMPUTATIONSMAIN$(\hat{V}_1(\gamma), \ldots, \hat{V}_n(\gamma), \gamma, h)$
4: **return** $\{\{(Q_t^{\gamma,h}, R_t^{\gamma,h})\}_{\gamma \in \Gamma, h \in H}\}_{t=1}^n$

---

**Lemma 12.** *The total query complexity of* PRECOMPUTATIONSFULL *is*

$$O(n \cdot \log(n) \cdot \log(k) \cdot u(n, k)).$$

*Proof.* For any $\gamma \in \Gamma$, let $n^\gamma$ be the length of the stream corresponding to predicted active elements $\hat{V}_1(\gamma), \ldots, \hat{V}_n(\gamma)$. By Lemma 7, the total query complexity is

$$\sum_{\gamma \in \Gamma} |H| n^\gamma u(n^\gamma, k) \leq \sum_{\gamma \in \Gamma} \log(n) \cdot 2 |\hat{V}(\gamma)| \cdot u(n, k)$$

$$= 2(\log n) u(n, k) \sum_{a \in \hat{V}} |\Gamma(a)|$$

$$= O(\log(n) \cdot u(n, k) \cdot n \cdot \log k). \quad \square$$

## D.2   The full UPDATESOL subroutine

UPDATESOLFULL takes as input the strongly robust sets $(Q_t^{\gamma,h}, R_t^{\gamma,h})$, for all guesses $\gamma \in \Gamma$ and $h \in H$. It maintains a data structure $\{(B^\gamma, A^\gamma, \eta^\gamma)\}_{\gamma \in \Gamma}$ where $(B^\gamma, A^\gamma, \eta^\gamma)$ is the data structure maintained by UPDATESOLMAIN over guess $\gamma$ for OPT. UPDATESOLMAIN is called over all guesses $\gamma \in \Gamma$ such that there is at least one active element $a \in V_t(\gamma)$. The precomputations given as input to UPDATESOLMAIN with guess $\gamma \in \Gamma$ are $(Q_t^{\gamma,\eta_t'}, R_t^{\gamma,\eta_t'})$ where $\eta_t' \in H$ is the closest guess in $H$ to the current prediction error $\eta_t$ that has value at least $\eta_t$. Note that $\eta_t'$ is such that $\eta_t' + 2w \geq k$ due to the definition of $H$. The solution returned at time $t$ by UPDATESOLFULL is the best solution found by UPDATESOLMAIN over all guesses $\gamma \in \Gamma$.

---

**Algorithm 7** UPDATESOLFULL

---

**Input:** function $f : 2^V \rightarrow \mathbb{R}$, data structure $A$, constraint $k$, precomputations $P_t = \{(Q_t^{\gamma,\eta}, R_t^{\gamma,\eta})\}_{\gamma \in \Gamma, \eta \in H}$, time $t$, current prediction error $\eta_t$, $V_t^1, V_t^2, \hat{V}_t$

1: $\{A^\gamma\}_{\gamma \in \Gamma} \leftarrow A$
2: $\eta_t' \leftarrow \min\{h \in H : h \geq \eta_t\}$
3: **for** $\gamma \in \Gamma$ such that $|V_t(\gamma)| > 0$ **do**
4:     $A^\gamma, S^\gamma \leftarrow$ UPDATESOLMAIN$(f, A^\gamma, k, (Q_t^{\gamma,\eta_t'}, R_t^{\gamma,\eta_t'}), t, \eta_t', V_t^1(\gamma), V_t^2(\gamma), V_{t-1}(\gamma), \gamma)$
5: $A \leftarrow \{A^\gamma\}_{\gamma \in \Gamma}$
6: $S \leftarrow \text{argmax}_{\gamma \in \Gamma : |V_t(\gamma)| > 0} f(S^\gamma)$
7: **return** $A, S$

---

**Lemma 13.** *The* UPDATESOLFULL *algorithm has, over the $n$ time-steps of the streaming phase, an amortized query complexity per update of $O\left(\log^2(k) \cdot u(\eta + w + k, k)\right)$.*

*Proof.* For every $\gamma \in \Gamma$, we consider the following stream associated to $\gamma$: $\{(a_t', o_t')\}_{t=1}^n$ where $(a_t', o_t') = (a_t, o_t)$ if $\gamma \in \Gamma(a_t)$ and $(a_t', o_t') = (\texttt{null}, \texttt{null})$ otherwise. $\texttt{null}$ is a dummy symbol denoting an absence of an insertion or deletion. Let $n^\gamma$ be the number of insertion/deletions in this new stream, i.e., $n^\gamma = \sum_{t=1}^n \mathbb{1}[a_t \neq \texttt{null}]$.

Using Lemma 5 the total query complexity due to a single phase corresponding to a $(\gamma, \eta^\gamma)$ pair is $O(u((\eta^\gamma + w + k) \log k, k) \cdot (\eta^\gamma + w + k) \log k)$. Next, note that there is a one-to-one mapping between insertions/deletions to $A^\gamma$ at line 9 or line 11 of UPDATESOL and elements of the stream

$\{(a'_t, o'_t)\}_{t=1}^n$. Thus, since a phase ends when $|\text{Ops}^\star(A)| > \eta^\gamma/2 + w$, all phases except the last phase process at least $\eta^\gamma/2 + w$ non-null elements from the stream $\{(a'_t, o'_t)\}_{t=1}^n$. Thus, if there are at least two phases corresponding to $n^\gamma$ then the total query complexity over these $\eta^\gamma/2 + w$ non-null elements is $O(u(\eta^\gamma + w + k, k) \cdot (\eta^\gamma + w + k) \log k \cdot \frac{n^\gamma}{\eta^\gamma/2+w})$. Since, $\eta^\gamma/2 + w > k$ we have that the amortized query complexity over $n^\gamma$ non-null elements is $O(u(\eta^\gamma + w + k, k) \cdot \log k)$. We also have that $O(u(\eta^\gamma + w + k, k) \cdot \log k) = O(u(\eta + w + k, k) \cdot \log k)$ because either $\eta + 2w > k$ which implies that $\eta^\gamma + w = O(\eta + w)$ or $\eta + 2w < k$ in which case $\eta^\gamma + w + k = O(k)$.

If there is only one phase, i.e. when $n^\gamma \le \eta^\gamma/2 + w$, then the amortized query complexity for the first phase is upper bounded by $u(n^\gamma, k) = O(u(\eta^\gamma/2 + w, k))$. Thus, the amortized query complexity due to $\gamma$ is $O(u(\eta + w + k, k) \log k)$, for any $\gamma \in \Gamma$. We get that the total query complexity is

$$\sum_{\gamma \in \Gamma} O(u(\eta + w + k, k) \cdot n^\gamma \cdot \log k) = \sum_{\gamma \in \Gamma} \sum_{a \in \Gamma(a)} O(u(\eta + w + k, k) \log k)$$
$$= \sum_{a \in V} \sum_{\gamma \in \Gamma(a)} O(u(\eta + w + k, k) \log k)$$
$$= O(n \log^2 k) \cdot u(\eta + w + k, k) \quad \square$$

### D.3 The main result

By combining the algorithmic framework (Algorithm 1) together with subroutines UPDATESOLFULL and PRECOMPUTATIONSFULL, we obtain our main result.

**Theorem 2.** *Algorithm 1 with subroutines* UPDATESOLFULL *and* PRECOMPUTATIONSFULL *is a dynamic algorithm that, for any tolerance $w$ and constant $\epsilon > 0$, achieves an amortized expected query complexity per update during the streaming phase of $O(\text{poly}(\log \eta, \log w, \log k))$, an approximation of $1/2 - \epsilon$ in expectation, and a query complexity[7] of $\tilde{O}(n)$ during the precomputation phase.*

*Proof.* The dynamic algorithm DYNAMIC used by the PRECOMPUTATIONS and UPDATESOL subroutines is the algorithm of Lattanzi et al. [20] with amortized expected update time $u(n, k) = O(\text{poly}(\log n, \log k))$. By Lemma 13, the amortized expected query complexity is

$$O\left(\log^2(k) \cdot u(\eta + 2w + k, k)\right) = O\left(\text{poly}(\log(\eta + w + k), \log k)\right).$$

For the approximation, consider some arbitrary time $t$. Let $\gamma^\star = \max\{\gamma \in \Gamma : \gamma \le (1 - \epsilon)\text{OPT}_t\}$. Let $\text{OPT}'_t := \max_{S \subseteq V_t(\gamma^\star):|S| \le k} f(S)$. We have that

$$\text{OPT}'_t \ge f(O_t \cap V_t(\gamma^\star)) \ge_{(1)} f(O_t) - \sum_{o \in O_t \cap V_t(\gamma^\star)} f(o) \ge_{(2)} f(O_t) - k \cdot \frac{\epsilon \gamma^\star}{k} \ge_{(3)} (1 - \epsilon)\text{OPT}_t$$

where (1) is by submodularity, (2) is by definition of $V_t(\gamma^\star)$, and (3) by definition of $\gamma^\star$. Consider the calls to UPDATESOL by UPDATESOLFULL with $\gamma = \gamma^\star$. Let $t'$ be the time at which $A_t^{\gamma^\star}$ was initialized by UPDATESOL with precomputations $(Q_{t'}, R_{t'}) = (Q_{t'}^{\gamma^\star, \eta'_{t'}}, R_{t'}^{\gamma^\star, \eta'_{t'}})$, where this equality is by definition of UPDATESOLFULL.

Next, we show that the conditions to apply Lemma 4 are satisfied. By definition of PRECOMPUTATIONSFULL and Lemma 6, $(Q_{t'}, R_{t'})$ are $(d = 2(\eta'_{t'} + 2w), \epsilon, \gamma^\star)$-strongly robust with respect to $V_t(\gamma^\star)$ with $\gamma^\star = \gamma_{t'}$. Since $\eta'_{t'} \ge \eta_{t'}$, we have that $(Q_{t'}, R_{t'})$ are $(d = 2(\eta_{\text{old}} + 2w), \epsilon, \gamma^\star)$. We also have that

$$\gamma_{t'} \le (1 - \epsilon)\text{OPT}_t \le \text{OPT}'_t \le \text{OPT}_t \le (1 + \epsilon)\gamma_{t'}/(1 - \epsilon).$$

---

[7]We note that, despite the amortized query complexity of Lattanzi et al. [20] being in expectation, the asymptotic bound on the precomputation query complexity can hold deterministically, instead of in expectation, by forcing PRECOMPUTATIONSFULL to terminate if it has performed a number of queries that is larger than $\epsilon^{-1}$ times its expected number of queries (note that the precomputation query complexity only depends on known parameters, $n$ and $k$). By Markov's inequality, such an early termination happens with probability at most $\epsilon$. Thus, even with no guarantees on the approximation achieved in these early termination cases, the loss in the expected approximation caused by this forced termination is at most $1 - \epsilon$.

Thus, with $\epsilon' > 0$ such that $(1 + \epsilon') = (1 + \epsilon)/(1 - \epsilon)$, we get $\gamma_{t'} \leq \text{OPT}'_t \leq (1 + \epsilon')\gamma_{t'}$. By Lemma 3, DYNAMIC is a threshold-based algorithm. Thus, all the conditions of Lemma 4 are satisfied and we get that the solution $S_t^{\gamma^\star}$ returned by the call to UPDATESOL with $\gamma = \gamma^\star$ at time $t$, where $A_t^{\gamma^\star}$ was initialized by UPDATESOL at time $t'$ with $\gamma = \gamma^\star$, is such that $\mathbf{E}[f(S_t^{\gamma^\star})] \geq \frac{1-5\epsilon'}{2}\gamma^\star \geq (1 - \epsilon)\frac{1-5\epsilon'}{2}\text{OPT}_t$. Finally, since UPDATESOLFULL returns, among all the solutions returned by UPDATESOL, the one with highest value, it returns a set $S_t$ such that $\mathbf{E}[f(S_t)] \geq \mathbf{E}[f(S_t^{\gamma^\star})] \geq (1 - \epsilon)\frac{1-5\epsilon'}{2}\text{OPT}_t$. Finally, the precomputation query complexity is by Lemma 12 with $u(n, k) = O(\text{poly}(\log n, \log k))$. $\qquad\square$

