# OpenReview forum: "Learning-Augmented Dynamic Submodular Maximization"
_NeurIPS.cc/2024/Conference — NeurIPS 2024 poster_

### Official Review · Reviewer_a2JE · 2024-07-12

**Soundness:** 3
**Presentation:** 3
**Contribution:** 3
**Rating:** 7
**Confidence:** 3

**Summary:**

Authors consider submodular maximization with cardinality constraint
in dynamic setting: Algorithm sees a series of $n$ insertions and deletions
of elements and has to maintain a subset of active elements
maximizing given submodular function.
Best known algorithms for this problem by Lattanzi et al., Monemizadeh,
and Banihashem et al., achieve approximation ratio 0.5-eps
with amortized update time polylogarithmic in $n$ and $k$, where
$n$ is the length of the input stream and $k$ denotes the cardinality
constraint.

In this paper, authors propose an algorithm which receives predictions
about insertion and deletion time of each element beforehand.
This allows the algorithm to precompute an update strategy assuming that the
predictions are correct.
Given a parameter $w$, authors define the prediction error $\eta$ as the number
of elements whose predicted insertion or deletion time is not within
$w$ time steps from the real one.
Their main result is an algorithm with approximation ratio 0.5-eps
whose amortized update time is polylogarithmic in $\eta$, $w$, and $k$.
With $\eta = o(n)$, their algorithm achieves an assymptotic improvement
over the existing algorithms which do not use predictions.

**Strengths:**

* result seems strong and requires introduction of new ideas as well
as proving new properties about existing algorithms

**Weaknesses:**

* the predictions used are quite verbose:
having all predictions ahead of time is quite a restrictive requirement.
Authors pose utilization of predictions which come one by one as an open
problem.

**Questions:**

* where does dependence on epsilon appear in your bounds in Theorem?
* is the algorithm of Lattanzi the only existing algorithm which is
threshold-based, or is this property more common among existing techniques?

**Limitations:**

Clearly explained.

---

> ### Author Rebuttal · Authors · 2024-08-06
>
> We thank the reviewer for their comments!
>
> - *"where does dependence on epsilon appear in your bounds in Theorem?*"
>
> Our approximation is 1/2 - epsilon. In addition, the query complexity per update during the streaming phase and the query complexity during the precomputation phase both have a polynomial dependence on epsilon.
>
> - *"is the algorithm of Lattanzi the only existing algorithm which is threshold-based, or is this property more common among existing techniques?*"
>
> Algorithms that used threshold-based ideas are very common in submodular maximization. One of the first such algorithms is by Badanidiyuru-Vondrák in their 2013 paper on “Fast Algorithms for Maximizing Submodular Functions:.

---

> > ### Comment · Reviewer_a2JE · 2024-08-09
> >
> > thank you for your answers.

---

### Official Review · Reviewer_T4wq · 2024-07-12

**Soundness:** 3
**Presentation:** 4
**Contribution:** 3
**Rating:** 6
**Confidence:** 4

**Summary:**

The authors studied monotone submodular maximization under a cardinality constraint in a dynamic model where predictions of insertions and deletions are given. In submodular maximization, a ground set of elements and a function assign a value to any subset of these elements. A function is submodular if adding an element to a smaller set contributes more value than adding it to a larger set. A function is monotone if its value always increases with the set size. In the dynamic model, elements are inserted and removed, and only elements that are inserted but not deleted can be selected. The error $\eta$ is defined as the number of elements whose actual insertion or deletion time differs from the prediction by at least ww. The goal is to find, after each update, a subset of size at most k from the current elements that maximizes the value of the monotone submodular function while minimizing the number of query calls.

Previous works on dynamic monotone submodular maximization under cardinality constraints (without predictions) achieve a 1/2 approximation factor using O(polylog(n)) or O(k⋅polylog(k)) query calls. There are also works on dynamic monotone submodular maximization with predictions under matroid constraints, which is a generalization of cardinality constraints, achieving a ∼0.32 approximation factor. In this paper, the authors design an algorithm for dynamic monotone submodular maximization under cardinality constraints using predictions that achieve a 1/2 approximation factor with only $O(poly(log(\eta),log(w),log(k)))$. When the predictions are poor, the algorithm’s complexity is as bad as the algorithm without predictions, having O(polylog(n)) query complexity since $\eta$ would become $\theta(n)$.

They use precomputation to compute a solution based on predictions for each time and leverage previous works on the dynamic model and delete-robustness model to update their solution according to real insertions and deletions. They also run experiments, comparing their query complexity with the dynamic algorithm. The results show that when prediction errors are small, their algorithm requires significantly fewer query calls. However, when the predictions are poor, the query complexity is comparable to that of the dynamic algorithm.

**Strengths:**

Recently, many learning-augmented algorithms have been developed for dynamic and online models. They are interesting because, in real applications, we sometimes have an idea of how elements will change, raising the question of how we can improve our algorithms by leveraging these predictions. In this work, the authors improved the query complexity, eliminating the dependency on n and reducing the dependency on k to only logarithmically. It is interesting how query complexity can be improved using predictions, and it would be even more interesting if the approximation factor could also be improved.

**Weaknesses:**

There were different algorithms for this problem without using predictions: one using O(polylog(n)) (first algorithm) and the other using O(k⋅polylog(k)) (second algorithm) query calls. Since the authors' algorithm has polylog(k) in its query complexity, if k is as large as $\theta(n^\epsilon)$ where $\epsilon$ is a constant, their algorithm is not better than the first algorithm. If k is small, the second algorithm is not that bad. Still, in the second case, the authors' algorithm is slightly better, but we should note that they also have O(n⋅polylog(n)) for precomputation.

## Comments for the authors:
- Line 37: cited [30] twice
- Some sentences are too long and make it hard to read. For example, look at the sentences from lines 60 to 63, and the next sentence.

**Questions:**

I see that you mentioned Chen and Peng [10] proved that a dynamic algorithm for this problem needs poly(n)poly(n) query complexity, but does their approach apply when we have good predictions? I understand this question may be difficult and don’t expect the authors to answer it, but if they can answer affirmatively, it makes their result much stronger.

---

> ### Author Rebuttal · Authors · 2024-08-06
>
> We thank the reviewer for their comments!
>
> - *"Line 37: cited [30] twice"*
> - *"Some sentences are too long and make it hard to read. For example, look at the sentences from lines 60 to 63, and the next sentence."*
>
> Thank you, we have fixed the double citation and shortened these sentences to “A dynamic algorithm with predictions consists of two phases. During the precomputation
> phase at t = 0, the algorithm uses the predictions to perform queries before the start of the
> Stream. During the streaming phase at time steps t > 0, the algorithm performs queries, and uses the precomputations, to maintain a good solution with respect to the true stream. In this model, there is a trivial algorithm that achieves a constant update time when the predictions
> are exactly correct and an O(u) update time when the predictions are arbitrarily wrong. Here, u
> is the update time of an arbitrary algorithm A for the problem without predictions.“
>
>
> - *"I see that you mentioned Chen and Peng [10] proved that a dynamic algorithm for this problem needs poly(n) query complexity, but does their approach apply when we have good predictions? I understand this question may be difficult and don’t expect the authors to answer it, but if they can answer affirmatively, it makes their result much stronger."*
>
> The result from Chen and Peng that shows that poly(n) query complexity per update is necessary to achieve an approximation better than 1/2 does NOT hold when we have perfect predictions. In particular, it is possible to precompute solutions S_t that achieve a 1-1/e approximation for each time step during the precomputation phase and then use these solutions during the streaming phase while having 0 queries per update when the predictions are exactly correct. It is an interesting question for future work whether a 1-1/e approximation with an update time faster than poly(n) can be achieved not only when the predictions are exactly correct but also when the prediction error is small.

---

> > ### Comment · Reviewer_T4wq · 2024-08-09
> >
> > Thank you for your response. I don’t have any further concerns.

---

### Official Review · Reviewer_A7Fx · 2024-07-13

**Soundness:** 3
**Presentation:** 4
**Contribution:** 3
**Rating:** 8
**Confidence:** 4

**Summary:**

The paper studies the monotone dynamic submodular maximization problem under a cardinality constraint $k$ in the framework of algorithms with predictions. The authors consider a prediction model where the insert and delete times of the elements are predicted at time 0, and for any window size $w$, the prediction error $\eta$ is defined to be the number of elements whose actual insertion or deletion times differ from the predicted insertion or deletion times by more than $w$. Their main result is an algorithm that produces a $(1/2-\epsilon)$-approximate solution at each time step with expected amortized update time $O(\text{poly}(\log \eta, \log w, \log k))$ and preprocessing time $\tilde{O}(n)$.

**Strengths:**

* The problem studied is important and interesting to the NeurIPS community
* The result is strong, and the prediction model and the notion of error are natural. The algorithm is robust in the sense that it can handle an arbitrary number of elements with low prediction error (that fall within the $w$ window of their actual insertion/deletion time) and a reasonable number of elements with high error (whose predictions are off by more than $w$, counted by $\eta$). Moreover, their algorithm can handle elements that are not predicted to arrive but actually show up in the input sequence (these elements contribute to $\eta$). The performance of the algorithm also degrades gracefully as the prediction error increases.
* While the subject is inherently complicated as there are lots of parameters involved, the authors did a good job keeping everything clear and precise so that it is relatively easy to follow. The notation is also good. Moreover, the warm-up algorithm helps the reader understand some of the main ideas of the final algorithm.

**Weaknesses:**

* In the case where $k = o(\log n)$ and the prediction error $\eta = \Omega(n)$, the update time of the algorithm is worse than the update time $O(k \cdot \text{polylog}(k))$ achieved in reference [7]. Thus, in some cases, the algorithm performs worse than a worst-case algorithm without predictions.

**Questions:**

* Why is not the algorithm with update time $O(k \cdot \text{polylog}(k))$ presented in [7] compared to your learning-augmented algorithm theoretically or empirically?

**Limitations:**

The authors state their theoretical results formally, describing all assumptions.

---

> ### Author Rebuttal · Authors · 2024-08-06
>
> We thank the reviewer for their comments!
>
> - *“In the case where k=o(log⁡n) and the prediction error η=Ω(n), the update time of the algorithm is worse than the update time O(k⋅polylog(k)) achieved in reference [7]. Thus, in some cases, the algorithm performs worse than a worst-case algorithm without predictions.” and “Why is not the algorithm with update time O(k⋅polylog(k)) presented in [7] compared to your learning-augmented algorithm theoretically or empirically?”*
>
> The reviewer is correct that there are cases where our update time is worse than the update time O(k⋅polylog(k)). Our main goal was to achieve a worst-case update time that is at most the O(poly(log n, log k)) update time achieved by Lattanzi et al.. We believe that achieving a worst-case update time that is at most the O(k⋅polylog(k)) update time achieved by Banihashem et al., while also achieving an improved update time when the predictions are accurate, is an interesting open question. At the time when we performed the experiments, the paper by Banihashem et al. was not online yet. We plan on adding their algorithm as a benchmark for our experiments in the next version of the paper.

---

> > ### Comment · Reviewer_A7Fx · 2024-08-12
> >
> > Thank you for your response.

---

### Official Review · Reviewer_L8zN · 2024-07-15

**Soundness:** 3
**Presentation:** 3
**Contribution:** 3
**Rating:** 4
**Confidence:** 3

**Summary:**

This paper studied the dynamic submodular maximization problem with predictions. The goal of the problem is to maintain a high-quality solution in the presence of insertions and deletions. The main contribution is leveraging predictions, in the form of the pattern of insertions and deletions, to accelerate the update time of dynamic submodular maximization algorithms.

**Strengths:**

1. The paper is well-written and easy to read. The proposed solution and analysis, especially the connection to the robust submodular maximization problem, are technically sound.
2. The results improve upon existing ones when the prediction is accurate.

**Weaknesses:**

1. The paper does not provide enough details on how predictions are obtained in real life. For example, it lacks specific information about machine learning algorithms that could potentially be used to make the predictions. This is concerning because the current notation of prediction error is defined in a worst-case manner. If there exists one prediction with poor accuracy, it might dramatically hurt the results. To fully benefit from their results, a very robust and stable prediction algorithm is required.
2. The tightness of their results is unclear. As mentioned in the introduction, if the prediction is 100% accurate, there exists a trivial algorithm that requires constant update time. However, even with zero prediction error, their bound does not reduce to a constant.
3. A more natural benefit of predictions is achieving an enhanced approximation ratio. While the authors mention existing studies along this direction, it is worth exploring whether predictions can improve the approximation ratio, using the same update time as algorithms without predictions.
4. Generally, incorporating predictions often sacrifices worst-case performance, which is especially true if the prediction is inaccurate. It would be beneficial for the authors to comment on this, specifically addressing to what extent bad predictions might hurt the update time of their algorithms.
5. The last comment is more of a clarification question instead of a comment. In the dynamic setting, why not simply use the state-of-the-art offline algorithm, such as the classic greedy algorithm, to solve the maximization problem in each round? I understand that this might incur higher update time, but if we consider a special case of the dynamic setting where there is only one round, it seems that the update time cannot be lower than the state-of-the-art offline algorithm. Is there any assumption regarding the length of the time horizon, for example, the number of rounds must be at least O(n)?

**Questions:**

Please refer to the weakness section.

**Limitations:**

Yes.

---

> ### Author Rebuttal · Authors · 2024-08-06
>
> We thank the reviewer for their comments!
>
> - *“If there exists one prediction with poor accuracy, it might dramatically hurt the results.”*
>
> We believe that there might have been a misunderstanding about the prediction error. The prediction error is the number of elements whose predicted inserted and deletion times were not sufficiently accurate. In particular, if the predicted inserted and deletion times of one element are completely wrong, then this only increases the prediction error eta by one. Thus, it is not the case that “if there exists one prediction with poor accuracy, it might dramatically hurt the results."
>
>
> - *“The paper does not provide enough details on how predictions are obtained in real life.”*
>
> Regarding how predictions could be obtained, consider the product recommendation application where a platform only wants to display items that are in stock. Platforms often have accurate predictions about when an item will go from in-stock to out-of-stock (due to current inventory and rate of purchase) or from out-of-stock to in-stock (due to inventory information provided by seller).
>
>
> - *"The tightness of their results is unclear. As mentioned in the introduction, if the prediction is 100% accurate, there exists a trivial algorithm that requires constant update time. However, even with zero prediction error, their bound does not reduce to a constant."*
>
> The reviewer is correct that our update bound is not constant even when the prediction error is zero. With the following simple change, our algorithm achieves a constant update time when the predictions are exactly correct, while also maintaining its current guarantees: (1) as additional precomputations, also compute a predicted solution S_t for each time t assuming the predictions are exactly correct, (2) during the streaming phase, as long as the predictions are exactly correct, return the precomputed predicted solution S_t. At the first time step where the predictions are no longer exactly correct, switch to our main algorithm in the paper.
>
>
> - *"A more natural benefit of predictions is achieving an enhanced approximation ratio. While the authors mention existing studies along this direction, it is worth exploring whether predictions can improve the approximation ratio, using the same update time as algorithms without predictions."*
>
> We agree that using predictions to improve the approximation is also an interesting direction. We note that the best potential improvement we can hope to achieve with a polynomial-time algorithm is to go from 1/2 to 1-1/e, which is the best approximation achievable in the offline setting.
>
>
> - *"Generally, incorporating predictions often sacrifices worst-case performance, which is especially true if the prediction is inaccurate. It would be beneficial for the authors to comment on this, specifically addressing to what extent bad predictions might hurt the update time of their algorithms."*
>
> As mentioned in lines 86-87, even when the prediction error is arbitrarily large, the update time of our algorithm asymptotically matches the O(poly(log n, log k)) amortized expected query complexity from Lattanzi et al.. Thus, asymptotically, incorporating predictions does not cause our algorithm to sacrifice worst-case performance in comparison to this previous work.
>
>
> - *"The last comment is more of a clarification question instead of a comment. In the dynamic setting, why not simply use the state-of-the-art offline algorithm, such as the classic greedy algorithm, to solve the maximization problem in each round? I understand that this might incur higher update time, but if we consider a special case of the dynamic setting where there is only one round, it seems that the update time cannot be lower than the state-of-the-art offline algorithm. Is there any assumption regarding the length of the time horizon, for example, the number of rounds must be at least O(n)?"*
>
> There is no assumption regarding the length of the time horizon. As mentioned by the reviewer, simply using offline greedy in each round would lead to slower update time. We note that the notation n in dynamic submodular maximization refers to the length of the stream (and not the size of the ground set). We also note that, since it is assumed that at time t=0 there are no active elements, the total number of elements that can be active at any time t is at most t. If there is only one time step (which is what we believe the reviewer means by “only one round”), then there is at most one element that was inserted and the maximizing problem over one element is trivial.
>
>
> We believe that we have addressed all the reviewer’s concerns. If there remains any concern, we would be happy to answer those during the reviewers-authors discussion phase.

---

> > ### Author Response · Authors · 2024-08-12
> >
> > Dear reviewer L8zN,
> >
> > Thank you again for your helpful suggestions. Since the author-reviewer discussion period is almost over, we wanted to see if our response addressed all your concerns. We would be happy to answer any follow-up questions you might have.

---

### Decision · Program_Chairs · 2024-09-25

**Decision:**

Accept (poster)

**Comment:**

Most of the reviewers agreed that the paper makes a strong theoretical contribution to learning-augmented algorithms for submodular maximization. The problem studied is well motivated and the contribution is relevant to the community. The reviewers also noted several weaknesses of this work, notably that there are relevant parameter regimes where the algorithm performs worse than worst case algorithms without predictions. Additionally, although the prediction model is natural, one of the reviewers noted that the paper does not address how to obtain predictions in practice. Overall, the strengths of this paper outweigh its weaknesses.